# Broad-high operating temperature range and enhanced energy storage performances in lead-free ferroelectrics

Weichen Zhao [1], Diming Xu [1] ✉, Da Li [1], Max Avdeev [2], Hongmei Jing[3], Mengkang Xu[4], Yan Guo[1], Dier Shi[5], Tao Zhou[6], Wenfeng Liu[7], Dong Wang [8] ✉ & Di Zhou [1] ✉

The immense potential of lead-free dielectric capacitors in advanced electronic components and cutting-edge pulsed power systems has driven enormous investigations and evolutions heretofore. One of the significant challenges in lead-free dielectric ceramics for energy-storage applications is to optimize their comprehensive characteristics synergistically. Herein, guided by phase-field simulations along with rational composition-structure design, we conceive and fabricate lead-free $Bi_{0.5}Na_{0.5}TiO_3$-$Bi_{0.5}K_{0.5}TiO_3$-$Sr(Sc_{0.5}Nb_{0.5})O_3$ ternary solid-solution ceramics to establish an equitable system considering energy-storage performance, working temperature performance, and structural evolution. A giant $W_{rec}$ of 9.22 J cm$^{-3}$ and an ultra-high η ~ 96.3% are realized in the BNKT-20SSN ceramic by the adopted repeated rolling processing method. The state-of-the-art temperature ($W_{rec} \approx 8.46 \pm 0.35$ J cm$^{-3}$, η ≈ 96.4 ± 1.4%, 25–160 °C) and frequency stability performances at 500 kV cm$^{-1}$ are simultaneously achieved. This work demonstrates remarkable advances in the overall energy storage performance of lead-free bulk ceramics and inspires further attempts to achieve high-temperature energy storage properties.

With the continuous growth of the world population and the development of the global economy and society, worldwide energy demand keeps increasing at an alarming rate. Due to the desperate issues, it is vital to exploit a variety of clean and sustainable energy sources[1–4]. Compared with various current energy storage and conversion devices (e.g., lithium-ion batteries, supercapacitors, solid oxide fuel cells), electrostatic capacitors made of dielectric materials have attracted ever-increasing attention up till now owing to their benefits in terms of swift charging-discharging rates, ultrahigh power density, excellent thermal stability, and prolonged storage lifespan[5–8]. Nonetheless, the comparatively low recoverable energy storage density ($W_{rec}$) of current dielectric ceramic capacitors had significantly hindered their practical utilizations in sophisticated electronic components and forefront pulsed power systems. Accordingly, substantial efforts have been undertaken to synergistically boost the comprehensive energy storage characteristics of dielectric materials, especially lead-free

[1]Electronic Materials Research Laboratory & Multifunctional Materials and Structures, Key Laboratory of the Ministry of Education & International Center for Dielectric Research, School of Electronic Science and Engineering, Xi'an Jiaotong University, 710049 Xi'an, Shaanxi, China. [2]Australian Nuclear Science and Technology Organization, Lucas Heights 2234 NSW, Australia. [3]School of Physics and Information Technology, Shaanxi Normal University, 710062 Xi'an, Shaanxi, China. [4]State Key Laboratory for Strength and Vibration of Mechanical Structures, School of Aerospace, Xi'an Jiaotong University, 710049 Xi'an, Shaanxi, China. [5]Department of Chemistry, Zhejiang University, 310027 Hangzhou, Zhejiang, China. [6]School of Electronic and Information Engineering, Hangzhou Dianzi University, 310018 Hangzhou, Zhejiang, China. [7]State Key Laboratory of Electrical Insulation and Power Equipment, Xi'an Jiaotong University, 710049 Xi'an, Shaanxi, China. [8]Frontier Institute of Science and Technology and State Key Laboratory for Mechanical Behavior of Materials, Xi'an Jiaotong University, 710049 Xi'an, Shaanxi, China. ✉e-mail: diming.xu@xjtu.edu.cn; wang_dong1223@mail.xjtu.edu.cn; zhoudi1220@gmail.com

materials, to fulfill the pressing demands of electronic devices for integration, miniaturization, and environmental friendliness[9–13].

Currently, common-utilized dielectric capacitors developed for energy storage include thin films, polymer-based thick films, and ceramic materials[1,10,13–19]. Among the candidate dielectric materials, bulk ceramics usually have low dielectric losses, high-temperature stability, and excellent fatigue resistance, enabling them to be more suitable for applications in various operational situations, such as aerospace, hybrid electrical vehicles, and electromagnetic pulse systems. Lead-free ceramics with relaxation properties for energy storage applications, for instance, $BaTiO_3$ (BT)[20–22], $K_{0.5}Na_{0.5}NbO_3$ (KNN)[5,23,24], $NaNbO_3$ (NN)[11,25,26], $Bi_{0.5}K_{0.5}TiO_3$ (BKT)[8,27], and $Bi_{0.5}Na_{0.5}TiO_3$ (BNT)[17,28–30]-based ceramics, have been extensively investigated in past few years. BNT-based materials exhibit intrinsic large saturation polarization ($P_{max}$) and are attributed to hybridization between Bi $6s$ and O $2p$ orbitals[31], which surpasses commonly used lead-free relaxor ferroelectric (RFE) counterparts. In light of this, BNT-based systems have received substantial attention in the field of energy storage and have been recognized as one of the most prospective eco-friendly materials for advanced pulsed power applications[4,28,32]. Typical BNT-based binary or ternary solid solutions have been widely studied recently, including BNT-BT, BNT-NN, BNT-SrTiO₃ (BNT-ST), and BNT-BT-NN, etc[33–36]. In addition, Che et al. constructed a BNT-Ag($Nb_{0.5}Ta_{0.5}$)$O_3$ (ANT) ceramic combined with defect engineering and realized a high $W_{rec}$ of 6.6 J cm⁻³ at 510 kV cm⁻¹, albeit less success in energy efficiency (η less than 75%)[37]. Thus, there is still a manifest challenge in obtaining ultrahigh energy storage density while maintaining high efficiency over a broad operating temperature in BNT-based ceramics.

Herein, we rationally design an effective strategy to maintain high energy storage performance upon a wide working temperature range guided by the phase-field method. Specifically, the manipulations of polymorphic polar nanoregions (PNRs) by constructing morphotropic phase boundary (MPB) in $Bi_{0.5}Na_{0.5}TiO_3$-$Bi_{0.5}K_{0.5}TiO_3$ (BNKT) binary system and incorporated $Sr(Sc_{0.5}Nb_{0.5})O_3$ (SSN) allow us to establish an equitable system considering energy storage performance, working temperature performance, and structural evolution. A gigantic $W_{rec}$ of 9.22 J cm⁻³, and significantly enhanced energy efficiency η of 96.3% at an external electric field of 535 kV cm⁻¹ are realized in the BNKT-20SSN ceramic by the adopted repeated rolling processing (RRP) method. Encouragingly, apart from the ultrahigh $W_{rec}$ and exceptional energy efficiency η mentioned above, remarkable temperature-insensitive performance ($W_{rec} \approx 8.46 \pm 0.35$ J cm⁻³, $\Delta\eta/\eta \leq 2\%$, 25–160 °C) and a slight fluctuation in frequency stability ($W_{rec} \approx 8.63 \pm 0.18$ J cm⁻³, $\Delta\eta/\eta < 3\%$, 1–100 Hz) at 500 kV cm⁻¹ are accomplished in the BNKT-20SSN ceramic (RRP). Simultaneously, further relationships among structural construction, energy storage performance, and working temperature performance are studied in-depth to elucidate the structure-performance connection upon temperature influence.

## Results

### Phase-field simulations of the structure construction process

The phase-field method is a powerful computational method to manifest the spatiotemporal evolution of microstructures and the related physical properties. The method is extensively and popularly utilized in dielectric materials, especially energy storage-related dielectric materials, to simulate the processes in terms of grain growth, solidification, thin-film deposition, crack propagation, and breakdown[5,38,39]. Here, we take $Bi_{0.5}Na_{0.5}TiO_3$-$Bi_{0.5}K_{0.5}TiO_3$ (BNT-BKT) binary system at the morphotropic phase boundary (MPB) as the initial phase and $Sr(Sc_{0.5}Nb_{0.5})O_3$ (SSN) as the guest phase to perform a two-dimensional phase-field simulation. The physical parameters and calculation parameters were set based on the 0.92BNT-0.18BKT (BNKT) system where rich polymorphic polar nanoregions (PNRs) could be observed over the coexistences of the rhombohedral (R3c) phase and the tetragonal (P4bm) phase.

The phase-field simulation results of the phase distribution as a function of the SSN phase fraction (see Fig. 1a) show that the R phase dominates in the BNKT end, albeit considerable T phase and local polar inhomogeneity exist. We employed a color scheme using RGB colors alongside vectors to depict different polarization orientations in the five left columns (vector contours). The colors visually distinguish ferroelectric domains characterized by distinct polar directions. However, distinguishing between different R values proved challenging due to the presence of eight <111> directions. To address this issue, in the right-most column (symmetry contours), we incorporated a color bar to depict the phase contour. It is essential to note that the organization of the color correspondence between these two types of plots is unrelated and follows different schemes. With the increase in the fraction of the guest SSN phase, the fraction of the T phase dramatically increases and the phase fraction reverses at around $x = 0.3$. Subsequently, the polarization distribution of the external electric field as a function of the SSN fraction was calculated, as also shown in Fig. 1a. Obviously, the long-range ferroelectric (FE) order of the host phase could be disrupted, and more abundant nanodomains are produced when $x$ increases, permitting the material to have the relaxor behavior[40]. The polarization directions are more disordered with less guest phase in the initial state, but it is harder to polarize fully. When the external electric field is applied and removed, the polarization exhibits a similar arranging behavior and restores its original state at the end, proving the process is reversible. We finally selected the BNKT-20SSN ($x = 0.2$) phase as the target composition, notwithstanding BNKT-30SSN ($x = 0.3$) presents a larger polarization response because the BNKT-20SSN phase displays a relatively reasonable structural construction and a considerable polar order–disorder arrangement as a relaxor ferroelectric. The initial state polarization distribution of the BNKT-20SSN phase to temperature, see Fig. 1b, strongly indicates that temperature is irrelevant to the constructed structure and could less infect the local energy barrier. The calculated polarization response to an external electric field (along the [100] axis) of the BNKT-20SSN ($x = 0.2$) phase is displayed in Fig. 1c. The calculated $P$–$E$ loop illustrates the polar structure evolution ($O_1$–$P_1$–$N$–$P_2$–$O_2$) responding quickly to an external electric field; the polarization distributions are close between $O_1$ and $O_2$ strongly prove the PNRs are reversible, resulting in a low remanent polarization ($P_r$).

### Energy storage performance, stability, and charge/discharge properties for practical application

Based on the phase-field simulation results above, we selected BNKT-20SSN as the target material for further study. Calcined BNKT-20SSN powders were used as the initial materials for preparing ceramic tapes by repeated rolling processing (RRP), which was employed to generate a dense structure with minimal porosity during sintering, enhancing the breakdown strength ($E_b$). As expected, the average grain size of the BNKT-20SSN ceramic (RRP) with compact microstructure is substantially finer than that of the sample prepared by conventional cold isostatic pressing (Supplementary Fig. 1a, b). Correspondingly, as presented in Supplementary Fig. 2, the theoretical $E_b$ of the BNKT-20SSN ceramic (RRP) evaluated by the Weibull distribution experiments is much higher than that of the cold isostatic pressing sample, and the Weibull modulus $\beta$ value of 19 (>10) was obtained, indicating the reliability of the breakdown strength[34]. The magnitude of $E_b$ is an essential factor affecting the energy storage density of dielectric materials. Figure 2a shows the room-temperature $P$–$E$ hysteresis loops of the BNKT-20SSN ceramic (RRP) measured from 100 kV cm⁻¹ to the critical electric field (535 kV cm⁻¹) at 10 Hz. Of particular noteworthy is that the $P$–$E$ loops show a near-zero $P_r$ and negligible hysteresis during the charging/discharging process, even at a high electric field of 535 kV cm⁻¹. In addition, the $I$–$E$ curve reveals that the initial broad peak ($P_1$) corresponds to the transformation of PNRs into a highly polarized state, while the subsequent peak ($P_2$) signifies the relaxation of the highly polarized state, see Supplementary Fig. 3. The current value

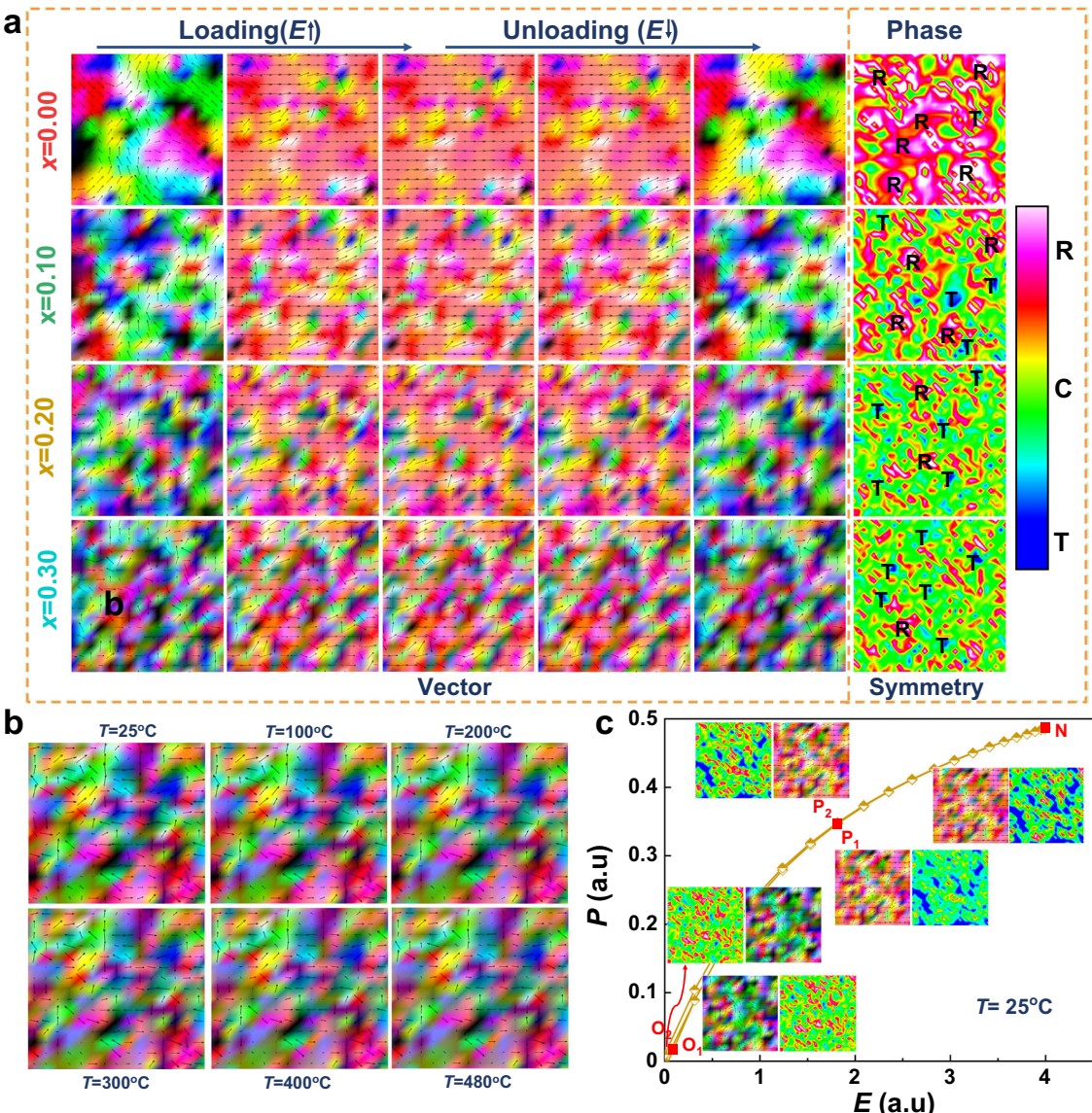

**Fig. 1 | The phase-field simulation results of (1-x)BNKT-xSSN. a** Calculated microstructural evolutions under given fields with vector contours and phase distributions with symmetry contours. **b** Magnification into the microstructure evolution of the BNKT-20SSN ceramic at various temperatures. **c** Calculated $P$–$E$ hysteresis loop and microstructure evolution of the BNKT-20SSN ceramic at 25 °C. Arrows capture the polarization magnitude and orientation.

remains unchanged from its original value upon unloading the positive field, indicating the absence of an irreversible component in the field-induced phase transitions of BNKT-20SSN ceramic[41,42]. As can be observed from Supplementary Fig. 4, both the $P_{max}$ and $\Delta P$ increase rapidly as the electric field rises. However, the $P_r$ value remains almost constant around zero even at the critical electric field, facilitating energy storage efficiency. As a result, a giant $W_{rec}$ (9.22 J cm$^{-3}$) and a prominent high-level of $\eta$ (96.3%) >95% are simultaneously achieved in the BNKT-20SSN ceramic (RRP) under the critical electric field of 535 kV cm$^{-1}$, which outperform the majority of recently reported lead-free bulk ceramics with admirable energy storage performance, as shown in Supplementary Fig. 5. Moreover, it was commendable that the BNKT-20SSN ceramic (RRP) demonstrates an ultrahigh energy storage performance at relatively high temperatures (-150 °C), surpassing the majority of lead-free bulk ceramics and even certain MLCCs (Multi-Layer Ceramic Capacitors), see Fig. 2b[5,6,8,9,17,20,23,28,29,40,43–53] and Supplementary Table 1. This achievement signifies the substantial potential of BNKT-20SSN ceramic (RRP) as a promising candidate for advanced high-temperature energy storage applications.

Apart from the ultrahigh $W_{rec}$ and $\eta$, another crucial criterion to determine the availability of high-power pulsed electronic components is the charge/discharge performance. The overdamped discharge property for the BNKT-20SSN ceramic (RRP) under various applied electric fields was measured at room temperature using a purpose-constructed resistance–inductance–capacitance (RLC) load circuit, and the regular overdamped oscillation waveforms reflect steady discharge behavior, as illustrated in Supplementary Fig. 6. Meanwhile, it can be observed that the discharge current rapidly approaches its peak ($I_{max}$ = 8.2 A, at 500 kV cm$^{-1}$) and then tends to diminish swiftly. Moreover, the high discharge energy density ($W_d$) -5.2 J cm$^{-3}$ can be liberated in a short period of time ($t_{0.9}$, 90% of $W_d$ is released) -244 ns at 500 kV cm$^{-1}$ (Fig. 2c). According to the foregoing data, the BNKT-20SSN ceramic (RRP) exhibits excellent charge/discharge characteristics, making it a promising candidate for pulsed power applications.

To ensure steady functioning in complex settings, ideal frequency reliability, and temperature stability must be guaranteed. The frequency-dependent energy storage property of BNKT-20SSN

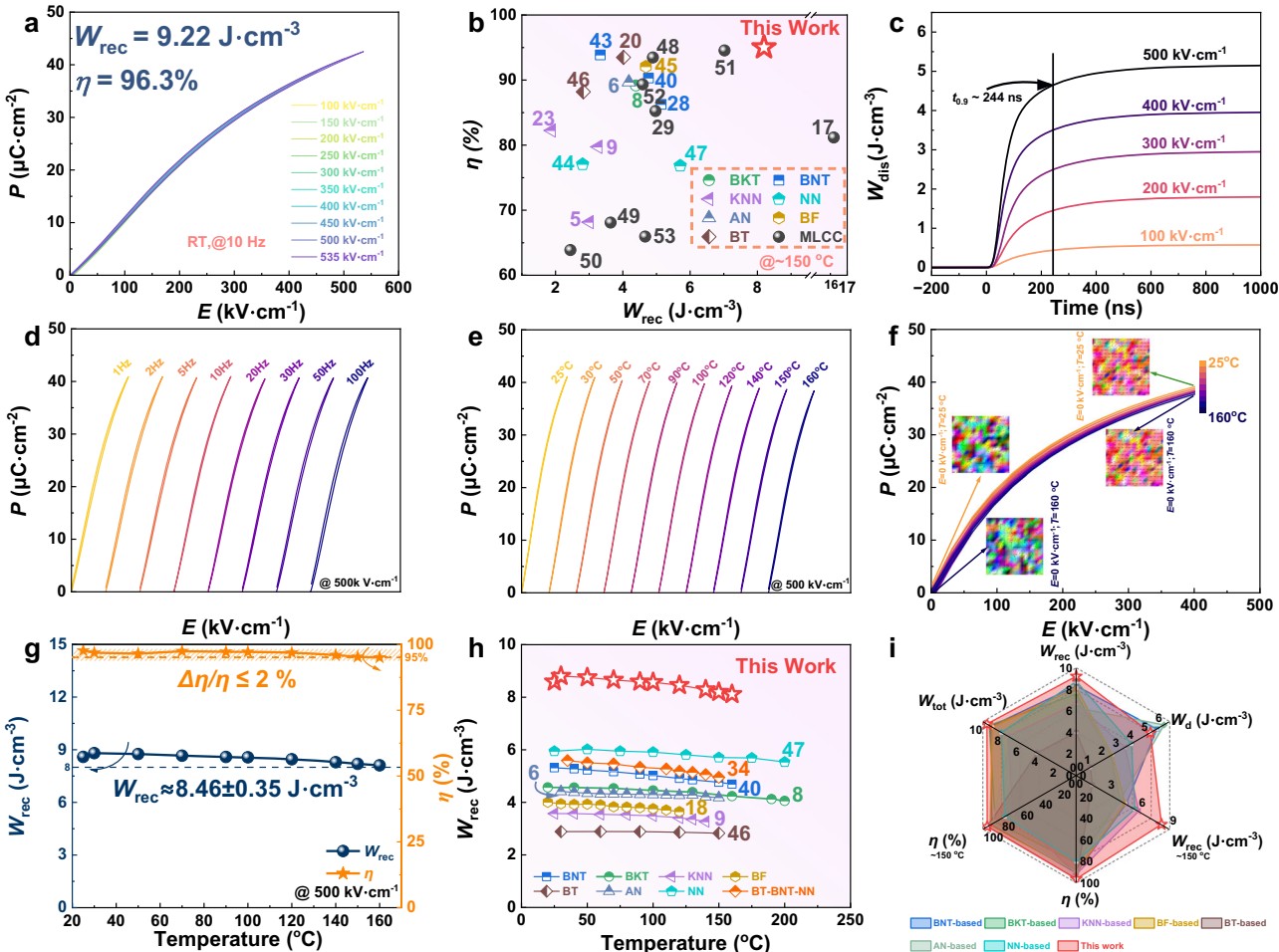

**Fig. 2 | The energy storage performance under various conditions and charge/discharge characteristics of BNKT-20SSN ceramic (RRP). a** Room-temperature $P$–$E$ loops measured till the critical electric field of the BNKT-20SSN ceramic (RRP). **b** Comparisons of $W_{rec}$ versus $\eta$ (-150 °C) between our work with some recently reported lead-free bulk ceramics and certain MLCCs (Multi-Layer Ceramic Capacitors). **c** Time dependence of discharge energy density under different electric fields ($R = 202\,\Omega$). **d** Frequency-dependent, and **e** temperature-dependent $P$–$E$ loops at 500 kV cm⁻¹. **f** Calculated $P$–$E$ loops and microstructure evolution as a function of $E$ at different temperatures. **g** $W_{rec}$ and $\eta$ as a function of temperature under 500 kV cm⁻¹. **h** A comparison of energy storage performances across a wide operating temperature range between our study and other reported bulk ceramics. **i** Comparisons of comprehensive properties ($W_{tot}$, $W_{rec}$, $\eta$, $W_d$, $W_{rec}$ -150 °C and $\eta$ -150 °C) between our study and other representative ceramics with excellent energy storage comprehensive performance.

ceramic (RRP) was investigated at 500 kV cm⁻¹, as shown in Fig. 2d, all $P$–$E$ hysteresis loops are slender with practically unchanged $P_{max}$ values at various frequencies. The corresponding results are recorded in Supplementary Fig. 7, it can be seen that the BNKT-20SSN ceramic (RRP) exhibits excellent frequency reliability ($W_{rec} \approx 8.63 \pm 0.18\,\text{J cm}^{-3}$, $\eta \approx 94.5\% \pm 2.4\%$) across the entire frequency range (1–100 Hz). Likewise, Fig. 2e gives the temperature stability of BNKT-20SSN ceramic (RRP) at 500 kV cm⁻¹, with the attainable $P_{max}$ value marginally decreasing as the enhancement of the random electric field raises the reaction rate of the PNRs during heating. As expected, this result fits well with the $P$–$E$ curves derived from phase-field simulations, as presented in Fig. 2f. Simultaneously, the corresponding microstructural evolution in the existence of the external electric field at various temperatures demonstrates that the domain size and polarization strength decrease as temperature increases (Fig. 2f and Supplementary Fig. 8). Of particular significance is that the BNKT-20SSN ceramic (RRP) features not only a wide operating temperature range (25–160 °C) but also an unprecedently high $W_{rec}$ ($\approx 8.46 \pm 0.35\,\text{J cm}^{-3}$) and $\eta$ ($\approx 96.4 \pm 1.4\%$) in contrast to certain other cutting-edge lead-free ceramics with excellent temperature stability (Fig. 2g, h)[6,8,9,18,34,40,46,47]. Furthermore, the comparisons of comprehensive performances ($W_{tot}$, $W_{rec}$, $\eta$, $W_d$, $W_{rec}$ -150 °C and $\eta$ -150 °C) between our study and

recently reported lead-free systems are summarized in Fig. 2i[5,6,8,40,45,47,54]. It is clear that the BNKT-20SSN ceramic (RRP) covers a vast region of the radar diagram, suggesting that overall considerable improvements of energy storage performance, particularly $W_{rec}/\eta$ at high temperatures (around 150 °C), and excellent overdamped discharge properties can be achieved simultaneously in studied samples. Within the measured cycling numbers of 1–10⁴ at 300 kV cm⁻¹, the values of $P_{max}$ and $P_r$ maintain stability as well, see Supplementary Fig. 9. In light of this, the BNKT-20SSN ceramic (RRP) outperforms other state-of-the-art lead-free ceramics in terms of overall electrical characteristics, implying a significant potential for practical application in high-performance pulsed power capacitors.

## Elucidation of multiphase-nanoregion coexistence and local atomic polar displacement

Phase-field method calculation permits an efficient strategy for the subsequent sample preparation and optimization process. As indicated by the calculations, rich polymorphic polar nanoregions (PNRs) could be observed in the BNKT-20SSN system with the coexistence of the rhombohedral ($R3c$) phase and the tetragonal ($P4bm$) phase. High-resolution neutron powder diffraction (NPD) was then conducted to verify the two-phase coexistence. The Rietveld refinement over the

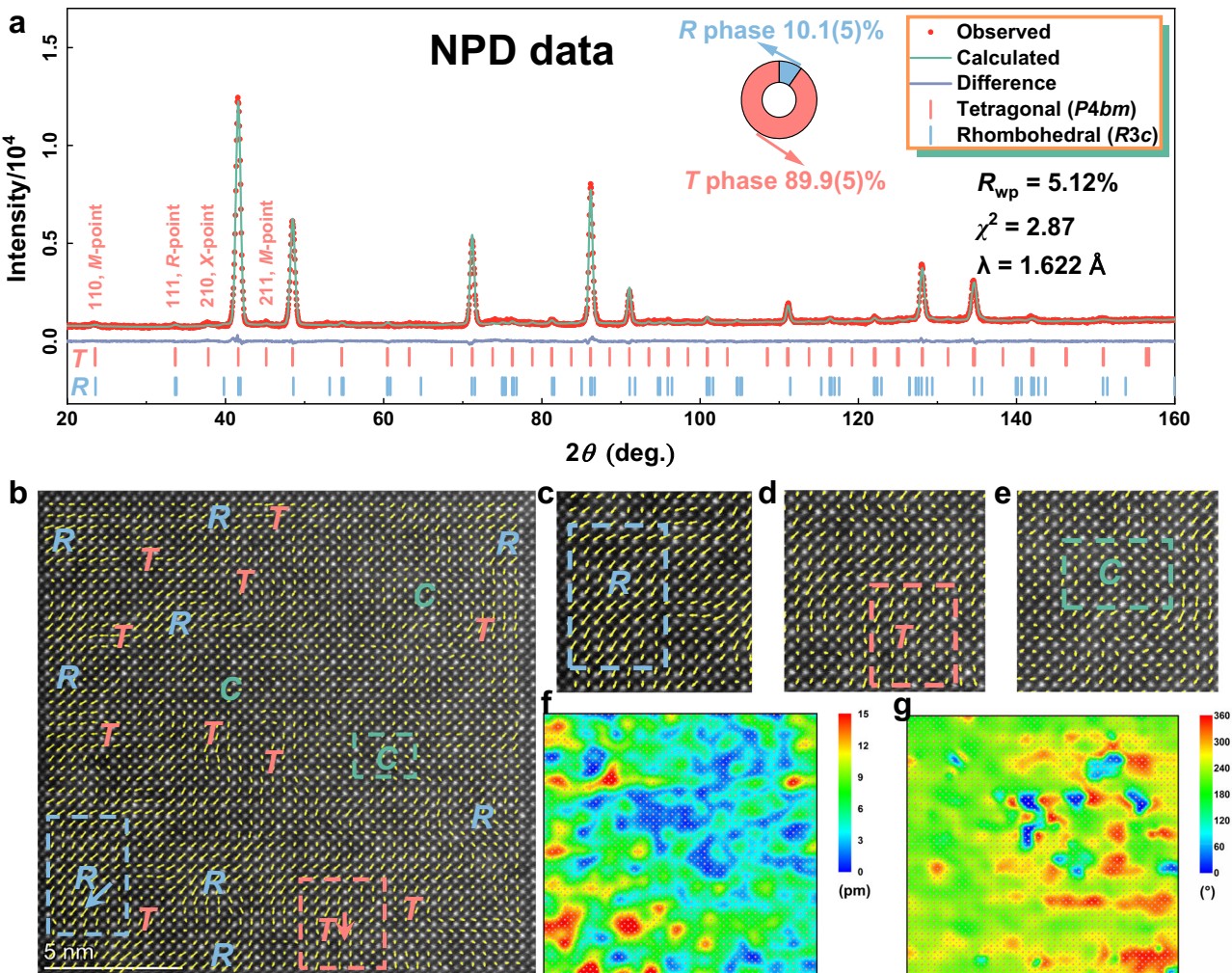

**Fig. 3 | Crystal structure analysis and local structure analysis of the BNKT-20SSN. a** Constant wavelength neutron powder diffraction refinement result at room temperature. **b** Atomic-resolution HAADF-STEM polarization vector image along [001] direction. **c**–**e** Magnification of the marked areas in (**b**). **f** Polarization magnitude mapping, and **g** polarization angle mapping.

NPD pattern gives a successful result, suggested by $\chi^2$ of 2.87 and $R_{wp}$ of 5.12%, and the fitted results of NPD data are presented in Fig. 3a and Supplementary Table 2[55]. Direct $R$ phase and $T$ phase fractions by the refinement are determined to be 10.1(5) wt% and 89.9(5) wt%, respectively, over the whole pattern. The atom fractions were set as constants in the refinement due to the value shifts dramatically in each refinement cycle, indicating that the cations in BNKT-20SSN form an ideal solid solution where $Sr^{2+}$ and $(Sc_{0.5}Nb_{0.5})^{4+}$ ions may totally diffuse into the matrix of the BNKT. Besides, no additional peaks could confirm the existence of a superlattice structure. Unlike the powder X-ray diffraction (PXRD) pattern, the NPD pattern is more sensitive to cations in the structure. The $M$-points (two odd numbers and one even number Miller indices), $R$-points (three odd numbers Miller indices, especially 111), and $X$-points (one odd number and two even numbers Miller indices) indicate the existence of in-plane tilting of the $T$ phase in the structure with the Glazer notion of $a^0a^0c^+$. Though all three indicator points could be observed in the NPD pattern, no peak splitting exists in either point. The weak intensities could thus come from the in-plane tilting by the $T$ phase, the local disorder-induced out-of-plane tilting by the cubic ($C$) phase, and/or cation-ordering over the A/B sites. However, the $X$-points are often the weakest intensities among the three indicator points, and the observed $X$-point shows a considerable intensity than the other two, particularly the $M$-points, suggesting the A-site disordering by the combination of $Bi^{3+}$, $Na^+$, $K^+$, and $Sr^{2+}$.

Atomic-resolution spherical aberration transmission electron microscopy was used in the scanning transmission electron microscopy (STEM) mode along with a high-angle annular dark-field (HAADF) detector in order to analyze the local structure of multiphase-nanoregion coexistence. Precise atom arrangement of the 100/010 plane was captured and fitted by the 2D Gaussian function, see Fig. 3b. The arrangement presents a direct process of the formation of polarization, the coexistence of multiphase PNRs, and nanoregion distribution. The 110-plane atom arrangement could hardly distinguish the $R$ and $T$ phases; the 111-plane atom arrangement shows only the overlap of A and B sites; the higher Miller indices planes projection images present insufficient resolution hindered by the Moiré fringe formation. With the processing by the 2D Gaussian function, the polarization vector could be straightforwardly described by a vector from the B-site cations center to the A-site cations corner, represented by the yellow arrows in Fig. 3b. The $R$ and $T$ phases could be directly identified by the long magnitude arrows, whereas the $C$ phase shows near-zero polarization. The bond lengths calculated from the B-site cations and the A-site cations along vertical and horizontal directions are noted as $c$ and $a$, and the $c/a$ ratio was then used to distinguish the $R$ phase and the $T$ phase, see Supplementary Fig. 10. Apparently, the captured image presents multiphase PNRs with $R$ and $T$ phases coexisting in the $C$ matrix. The size of the same polarization direction and magnitude PNRs is ~2 nm. The coexistent multiphase-nanoregion destroys the long-range ferroelectric

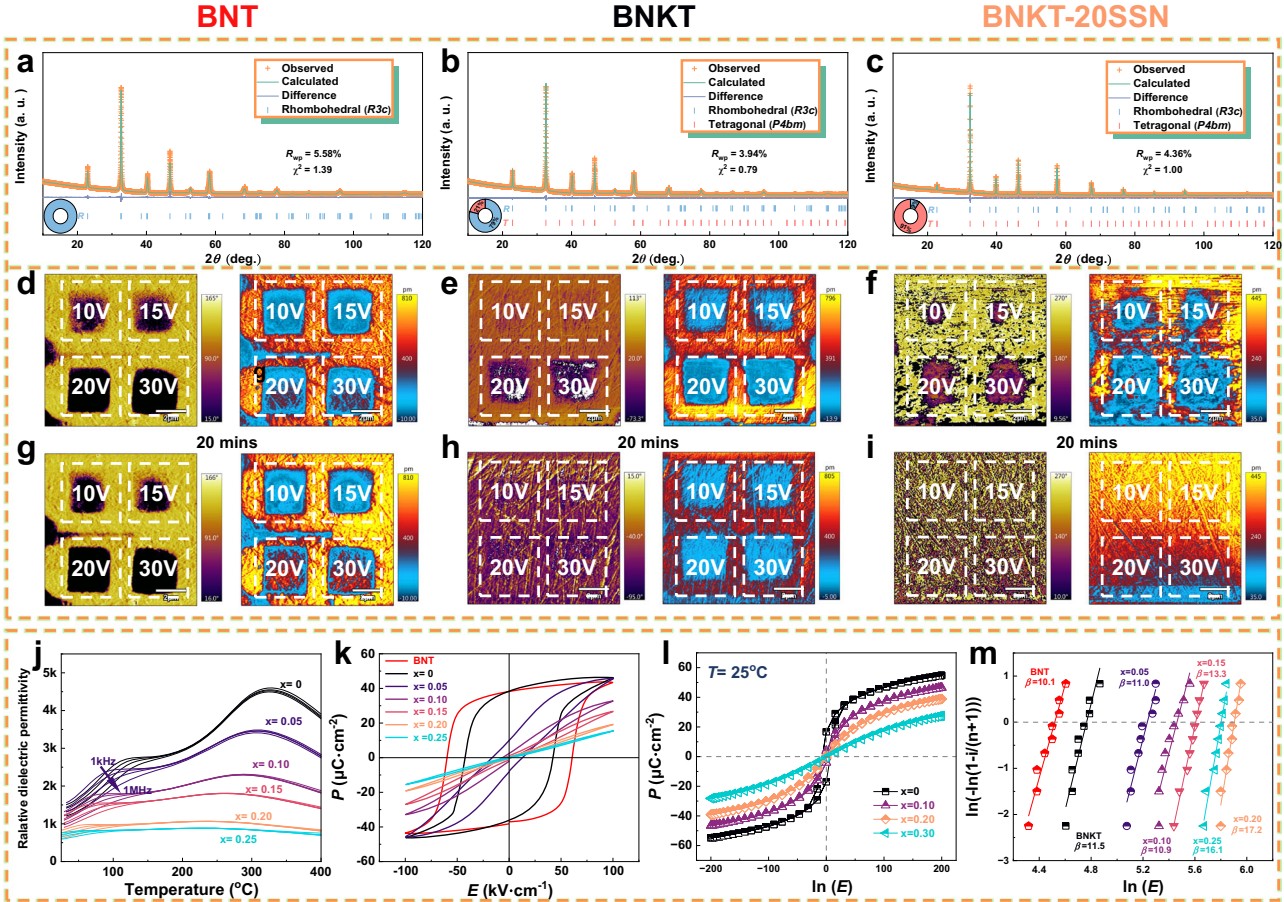

**Fig. 4 | Phase analysis, relaxation behavior, dielectric and ferroelectric properties of BNT and (1-x)BNKT-xSSN ceramics, along with their breakdown analysis.** The Rietveld refinement of PXRD patterns of **a** BNT, **b** BNKT, and **c** BNKT-20SSN. Out-of-plane PFM phase images along with amplitude after polarization with different voltages and relaxation durations, **d**, **g** BNT, **e**, **h** BNKT, and **f**, **i** BNKT-20SSN ceramics. **j** Temperature and frequency-dependent dielectric spectra. **k** Unipolar $P$–$E$ hysteresis loops at 100 kV cm⁻¹. **l** Calculated $P$–$E$ loops at 25 °C. **m** Weibull distribution of the breakdown strength on samples.

order and produces abundant small-size PNRs, consistent with those predicted by the phase-field calculation. Moreover, the randomly distributed nanoregions of different polarization directions and magnitudes could reduce polarization anisotropy. The enhanced polarization magnitude implies that the internal field is strong. The facts stated above could all lead to a more accessible and quicker polarization response to the external electric field, higher energy efficiency, and better energy storage performance. To better understand the polarizations in the sample, color-filled 2D patterns of summarized polarization directions and magnitudes are plotted in Fig. 3f, g. The patterns are highly corresponding to the phase-field calculation, signifying the reliability and the practical value of the phase-field prediction to design the dielectric materials for energy storage.

### Composition-driven and temperature-driven features

Pure BNT and (1-x)BNKT-xSSN ceramics were prepared to understand composition-driven and temperature-driven energy storage properties further. The powder X-ray diffraction (PXRD) patterns at room temperature (Supplementary Fig. 11a) show that all samples exhibit a typical BNT-based perovskite structure without impurities. The Rietveld refinement over PXRD patterns of each component was then processed by GSAS II software to determine the phases, and the results are displayed in Fig. 4a–c and Supplementary Fig. 12[55]. It is widely accepted that the BNT solid solution displays a rhombohedral ($R3c$) symmetry at room temperature[31], consistent with the refinement result. The ratio of the $R$ phase and the $T$ phase gradually decreases with the increase of $x$, and the pure $T$ phase was determined in the 0.82BNT-0.18BKT at 25SSN

ceramic, identical to those from previous studies[56–58]. Temperature-dependent PXRD patterns of the BNKT-20SSN in the temperature range of 25–200 °C (Supplementary Fig. 13) indicate no peak splitting or phase change occurs, suggesting good temperature stability. Furthermore, all elements exhibit homogeneous distribution characteristics with no segregation in BNKT-20SSN ceramic, see Supplementary Fig. 14.

The genesis of optimum energy storage properties is intimately connected to the dynamic reaction of domain structures to external electric fields[18,28]. Here, the dynamic domain response to an applied voltage and the relaxation behaviors of the BNT, BNKT, and BNKT-20SSN ceramics were characterized by employing piezo-response force microscopy (PFM), as displayed in Fig. 4d–i, the schematics of the applied voltage and region are similar to our previous work (Supplementary Fig. 15, 16)[40]. For the BNT ceramics (Fig. 4d, g), significant phase deviations and amplitude signals were observed even at a lower external voltage (10 V), and the overwhelming majority of domains did not flip after a 20 min relaxation time, demonstrating a robust FE characteristic with substantial $P_{max}$ and $P_r$. A similar occurrence was also seen in the BNKT ceramic (Fig. 4e, h), albeit the feedback signal (after 20 min) of the BNKT ceramic is slightly weaker than that of the BNT at an applied voltage of 10 V. On the contrary, the phase difference and amplitude signals of the BNKT-20SSN ceramic (Fig. 4f, i) are weak in the entire electrical scan zone and become more potent at higher external voltages. After a 20-min relaxation duration, the signals have significantly decreased. The facts mentioned above could all lead to a negligible $P_r$ and hysteresis loss in the BNKT-20SSN ceramic, thus increasing the energy storage efficiency.

In order to further investigate the mechanism of relaxation polarization behavior over a wide range of temperatures, the temperature (from room temperature to 400 °C) and frequency (from 1 kHz to 1 MHz) dependent dielectric permittivity of pure BNT and (1-$x$) BNKT-$x$SSN ceramics were performed and depicted in Fig. 4j and Supplementary Fig. 17. The pure BNT ceramic possesses two typical temperature-dependent dielectric anomalous peaks, labeled as "$T_s$" (depolarization temperature shoulder) and "$T_m$" (temperature of the maximum dielectric constant) at low and high temperatures, respectively[59–62]. At the ambient temperature ($T_{amb}$), as displayed in Supplementary Fig. 17a, the pure BNT ceramic exhibits a non-ergodic relaxor state and behaves as a normal ferroelectric with a $R3c$ symmetry space group[31,35,63]. Afterward, with the addition of BKT (Supplementary Fig. 17b), the BNKT ceramic with $R$ and $T$ phase coexistence at MPB displays a much higher maximum permittivity value (~4600 at 1 MHz) than that of the pure BNT ceramic (~2500 at 1 MHz). The $T_m$ of the BNKT ceramic shifts to a lower temperature, and a relatively broad permittivity peak can be detected in the MPB composition[64]. Subsequently, with increasing the content of SSN, the maximum value of the dielectric permittivity decreases step by step, see Fig. 4j. The two dielectric anomaly humps squint towards lower temperatures, and a gradually broaden temperature plateau of the dielectric constant can be observed. The analysis results of $\ln(1/\varepsilon' - 1/\varepsilon_m')$ versus $\ln(T - T_m)$ for BNT and (1-$x$)BNKT-$x$SSN ceramics are presented in Supplementary Fig. 18. It is observed that the calculated values of $\gamma$ for (1-$x$)BNKT-$x$SSN ceramics lie within the range of 1.86–1.99, indicating the pronounced relaxor behavior. Noteworthy, the dielectric relaxor or frequency dispersion behavior of the BNKT-20SSN ceramic could be noticed around $T_{amb}$, which allows for the formation of relaxor ferroelectric (a significant polar order–disorder arrangement) at room temperature and is beneficial to upgrading the energy storage performances[32].

The bipolar $P–E$ hysteresis loops of all components mentioned above were performed under 100 kV cm$^{-1}$ at room temperature, and the results are shown in Fig. 4k. The pure BNT ceramic exhibits a canonical FE behavior with large $P_{max}$, $P_r$, considerable hysteresis, and a fully saturated $P–E$ loop. A lower coercive electric field ($E_c$) and a slightly improved $P_{max}$ were obtained in the BNKT (MPB) ceramic, indicating a softened FE activity. The hysteresis losses decrease dramatically, accompanied by slimmer $P–E$ loops seen in the SSN-introduced compositions, and are consistent with those calculated from the phase-field simulation method (Fig. 4l). To better understand composition-driven breakdown strength, the values of theoretical $E_b$ of all ceramic samples were evaluated by the Weibull distribution experiments, see Fig. 4m. With increasing the SSN concentration, the theoretical $E_b$ values first increased and then decreased, reaching a maximum at the BNKT-20SSN ceramics. To investigate the energy storage performance of the ceramics, the $P–E$ hysteresis loops were performed on a ferroelectric analyzer. As depicted in Supplementary Fig. 19, the room-temperature $P–E$ loops of the BNKT-20SSN ceramic were measured from 100 kV cm$^{-1}$ to the critical electric field (385 kV cm$^{-1}$) at 10 Hz. As a consequence, a great $W_{rec}$ of 5.23 J cm$^{-3}$ along with a high energy efficiency of 90.2% are simultaneously achieved in the BNKT-20SSN ceramic, demonstrating a substantial promotional impact of combinatorial-optimization strategy guided by phase-field simulations on energy storage performance.

## Discussion

The selection of promising dielectric materials to combine is a foundational but inevitable procedure in material science. The presented phase-field method assistant strategy could be used to investigate dielectric material with high energy storage performance upon a wide working temperature range[5,40]. By learning the structural evolution, polarization distribution, and characteristics of PNRs in combinations of BNT-BKT-SSN at morphotropic phase boundary that affords possibly engineerable ceramic materials from the phase-field calculation

simulation strategy, we could initiatively decide and select the optimal composition for further study to achieve the remarkable performances.

To state, various combinations in perovskite structure have been considered to acquire substantial polar nanoregions in the structure to enhance energy storage performances. The initial BNT ceramic shows a considerable ferroelectric performance, but the $P_r$ is too large for energy storage applications[31,32,37]; following the BNKT binary system at the morphotropic phase boundary is an ideal fundament for massive afterward studies[56,61]. We intentionally choose ionic radius and valence distinguishable cation pairs as guests for targeting a solid solution and thus enable the discovery of diverse ferroelectric/energy storage performances. The phase-field method supports an efficient strategy of rational design of a targeted material with relaxor behavior by implying reduced PNRs, thermal stability, reversible response to the external electric field, high $P_{max}$, and low $P_r$, all of which could effectively optimize the ferroelectric/energy storage performances. The designed BNKT-20SSN ceramic (RRP) shows extraordinary performances of giant $W_{rec}$ and ultrahigh $\eta$, unprecedented temperature, and frequency stabilities. Multi-methods analysis describes the occurrence of the performances. The small, uniform, dense grain structures by the RRP process method guarantee the breakdown strength and subsequent applications. The coexisting $R/T/C$ multiphase nanoregions confirmed by atomic-scale TEM provide small-scaled PNRs with different polarization distributions in both magnitude and orientation, suggesting a relaxor-behavior FE with large $P_{max}$ of energy storage density and small $P_r$ of energy loss and allowing the polarization of nanoregions flipping along the same directions with lower energy and higher dynamics. The PNRs of the BNKT ceramic were confirmed with a size of ~200 nm, and the guest-invited BNKT-20SSN reduced the size to ~2 nm[65]. Notably, typical relaxor REs present only minor differences in polarization magnitude but only in orientation, which differs from what we reported here[1,66]. This phenomenon may be because of the considerable amount of SSN concentration, $x = 0.20$, and the fact that the A site is composed of 0.42Bi-0.34Na~0.08K~0.16 Sr, introducing considerably strong local heterogeneity and destabilizing the polarization ordering. The system retains the original PNRs of BNKT, but the PNRs are reduced and dissociated, forming a local order–disorder coexistence system. The metastable order–disorder system in BNKT-20SSN presents a lower $P_{max}$ than BNKT, but the abundant PNRs promote the performance of energy storage. Moreover, BNKT-20SSN ceramic (RRP) exhibits unprecedented temperature and frequency stabilities where the $W_{rec}$ and $\eta$ typically decrease when temperature increases, indicated by the decrease of $P_{max}$ and the occurrence of the non-negligible $P_r$. This case may come from a series of benefits. First, the average structure is stable upon temperature change to 200 °C. Second, the heterogeneity in the metastable order–disorder system may act as a pinning effect and hinder further atom thermal vibration-induced local structure change[67,68]. Finally, the randomly distributed PNRs also show some order–disorder, inviting a suitable local energy barrier to the external electric field and frequencies, indicated by the flat temperature-dependent dielectric response.

In summary, a rational phase-field calculation guided dielectric material designing method is presented to achieve small-scale and stable PNRs with overall well-behaved energy storage performance theoretically and experimentally. The so-calcined BNKT-20SSN guided by calculations with improved RRP techniques shows a $W_{rec}$ of 9.22 J cm$^{-3}$ and an ultrahigh $\eta$ of ~96.3% at large external electric field 535 kV cm$^{-1}$, accompanied with excellent temperature-insensitive ($W_{rec} \approx 8.46 \pm 0.35$ J cm$^{-3}$, $\eta \approx 96.4 \pm 1.4\%$, 25–160 °C) and frequency stability ($W_{rec} \approx 8.63 \pm 0.18$ J cm$^{-3}$, $\eta \approx 94.5\% \pm 2.4\%$, 1–100 Hz) at 500 kV cm$^{-1}$. This research provides a paradigm for the synergistic development of lead-free dielectric materials with enhanced comprehensive energy storage capacity over a broad operating temperature range to fulfill the pressing demands of modern energy storage components.

## Methods

### Ceramics preparation

The lead-free ceramics with the composition of $(1-x)$ $Bi_{0.5}(Na_{0.82}K_{0.18})_{0.5}TiO_3$-$xSr(Sc_{0.5}Nb_{0.5})O_3$ (abbreviated as BNKT-$x$SSN, where $x = 0$, 0.05, 0.10, 0.15, 0.20, and 0.25) were fabricated via a conventional solid-state reaction method. The pre-dried oxides and carbonate powders of $Bi_2O_3$ (≥99.9%), $Na_2CO_3$ (≥99.8%), $K_2CO_3$ (≥99.5%), $TiO_2$ (≥99.9%), $SrCO_3$ (≥99.5%), $Sc_2O_3$ (≥99.9%) and $Nb_2O_5$ (≥99.99%) (Aladdin Chemical Reagent Co., Ltd., CN) were used as the starting materials. The raw powders were weighed according to the stoichiometric ratio, and 2 wt.% excess of $Bi_2O_3$, $Na_2CO_3$, and $K_2CO_3$ were added to compensate for the weight loss at high temperatures. The mixtures were planetarily ball-milled for 24 h at 300 rpm with ethanol in nylon jars using Y-stabilized zirconia balls as milling media. After drying, the mixtures were calcined at 800–850 °C for 4 h in an alumina crucible and ball-milled again similarly. After drying and sieving, the powders were uniaxially pressed into pellets at 10 MPa in a 12 mm diameter stainless steel cylindrical die. Then, the preformed green ceramic sheets were pressed at 300 MPa for 3 min through cold isostatic pressing. Finally, the samples were sintered at 1140–1200 °C for 4 h with a heating rate of 4 °C min⁻¹ in closed alumina crucibles. To minimize the evaporation of the volatile elements Bi, Na, and K, the samples were embedded in powders of the same composition. For the electrical measurements, the silver paste was coated on both surfaces of the sintered samples and fired at 650 °C for 20 min to form electrodes.

### Repeated rolling processing

The green ceramic tapes with BNKT-20SSN composition were prepared by a repeated rolling processing (RRP) method, in which the pre-sintered powders were used as raw materials. After sieving, the calcined powders of BNKT-20SSN were thoroughly mixed with polyvinyl alcohol (PVA). The extrusion force generated by the two tightly contiguous rollers could greatly improve the density of the green ceramic tape and also help to form a dense structure during sintering with minimal porosity, resulting in a significant increase in $E_b$. And then, the ceramic tapes prepared by RRP were punched into disks with a diameter of 12 mm and then calcined at 650 °C for 2 h to burn out the PVA. Subsequently, the green disks were embedded in the presintered powders of the same composition and sintered at 1070–1140 °C for 2 h using a double crucible method, so that the volatilization of Bi, Na, and K elements during high-temperature sintering could be suppressed.

### Structural characterization

The phase purity and crystalline structure were characterized at an X-ray diffractometer (SmartLab-3 kW, Rigaku, Tokyo, Japan) with Cu Kα radiation. Neutron diffraction patterns were collected on the high-resolution powder diffractometer ECHIDNA at ANSTO over the angular range $8 ≤ 2\theta/° ≤ 160$, using a step size $\Delta 2\theta = 0.05°$ and a wavelength of 1.622 Å at room temperature. The Rietveld refinements on PXRD and NPD were analyzed using the program GSAS II[55]. The microstructure of the ceramics was observed by field-emission scanning electron microscopy (FE-SEM, FEI Quanta 250 FEG, Hillsboro, Oregon, USA). The Nano Measurer software was used to calculate the average grain size of the samples based on the SEM images. The HAADF atomic-scale images were acquired using an atomic-resolution STEM (aberration-corrected Titan Themis G2 microscope) and processed by 2D Gaussian fitting in MATLAB scripts to evaluate the polarization vector, magnitude, and angle maps.

### Dielectric measurements

For the dielectric measurements, the silver paste was coated on both surfaces of the sintered samples and fired at 650 °C for 20 min to form electrodes. Temperature and frequency-dependent dielectric

permittivity and tan$\delta$ were tested by an accurate inductor–capacitance–resistance (LCR) meter (E4980A, Agilent, USA) with a 3 °C min⁻¹ heating rate.

### Ferroelectric measurements

The room-temperature $P–E$ loops with a triangle signal of 10 Hz and temperature and frequency-dependent $P–E$ loops were measured on a ferroelectric analyzer (aix ACCT, TF Analyzer 2000, Aachen, Germany). For the $P–E$ loops measurements, the sintered disks were polished to ≈35 μm and then sputtered with gold electrodes on both surfaces.

### Pulsed charge–discharge test

The change–discharge properties of ceramics with a thickness of ≈50 μm were investigated via a specially designed resistance, inductance, and capacitance (RLC) load circuit.

### Dielectric breakdown test

The breakdown performances were measured via a ferroelectric analyzer (aix ACCT, TF Analyzer 2000, Aachen, Germany). The statistical $E_b$ properties for perovskite structure ceramics are generally estimated by Weibull distribution equations: $X_i = \ln(E_i)$, $Y_i = \ln(\ln(1/(1-P_i)))$, $P_i = i/(n+1)$, where $E_i$ is the experimental breakdown strength of each sample in ascending order, $P_i$ is the cumulative probability of dielectric breakdown, $n$ represents the total number of samples for ceramics. The Weibull distribution modulus, which can estimate the value of $E_b$, was represented by the slope of the fitting line.

### PFM characterization

The surface morphology, dynamic response, and relaxation behavior of the domains under external voltages were investigated by piezoelectric force microscopy (PFM, MFP-3D, Asylum Research, USA). The surfaces of each ceramic specimen used for PFM measurement were finely polished with polycrystalline diamond polishing paste.

### Phase-field simulations

Phase-field simulations have been employed to investigate a single crystal undergoing the Cubic (C) to Tetragonal (T) to Rhombohedral (R) ferroelectric transition. This study encompasses varying defect doping concentrations in the range of $x = 0$ to 0.30, denoted as BNKT-$x$SSN. The comprehensive evaluation of the ferroelectric system's total free energy is expressed as follows:

$$F = \int_V \left( f_{bulk} + f_{grad} + f_{couple} \right) dV + \int_V \left( f_{elas} + f_{elec} \right) dV \tag{1}$$

where $f_{bulk}$ represents the bulk free energy density,

$$
\begin{aligned}
f_{bulk} = {} & \alpha_1 \left( P_1^2 + P_2^2 + P_3^2 \right) - \alpha_{11} \left( P_1^2 + P_2^2 + P_3^2 \right)^2 + \alpha_{12} \left( P_1^2 P_2^2 + P_2^2 P_3^2 + P_1^2 P_3^2 \right) \\
& + \alpha_{112} \left( P_1^4 P_2^2 + P_2^4 P_3^2 + P_1^4 P_3^2 + P_1^2 P_2^4 + P_2^2 P_3^4 + P_1^2 P_3^4 \right) + \alpha_{113} \left( P_1^2 P_2^2 P_3^2 \right) \\
& + \alpha_{111} (P_1^2 + P_2^2 + P_3^2)^3
\end{aligned}
\tag{2}
$$

where $\alpha_{ij}$ represents the coefficient, the value of which is contingent upon both concentration $c$ and the temperature $T$.

$f_{grad}$ represents the gradient energy density.

$$
\begin{aligned}
f_{gradient} = {} & \frac{1}{2} G_{11}[(P_{1,1})^2 + (P_{1,2})^2 + (P_{1,3})^2 + (P_{2,1})^2 + (P_{2,2})^2 + (P_{2,3})^2 \\
& + (P_{3,1})^2 + (P_{3,2})^2 + (P_{3,3})^2]
\end{aligned}
\tag{3}
$$

where $G_{11}$ signifies the coefficient for gradient energy, while $f_{couple}$ pertains to the dipole effect induced by doping.

$f_{couple} = -\int d^3x \sum_{i=1,2,3} P_i(x) \cdot \varphi_{loc}(x)$, where $\varphi_{loc}(x)$ is the dipolar field generated through doping, expected to exhibit a random distribution and remain unchanged during the cooling process. $f_{elas}$ corresponds to the energy density associated with long-range elastic interactions, and $f_{elec}$ pertains to the energy density attributed to electrostatic interactions. $f_{elas} = \frac{1}{2}c_{ijkl}e_{ij}e_{kl} = \frac{1}{2}c_{ijkl}(\varepsilon_{ij} - \varepsilon_{ij}^0)(\varepsilon_{kl} - \varepsilon_{kl}^0)$, where $c_{ijkl}$ denotes the elastic constant tensor, $\varepsilon_{ij}$ signifies the total strain, and $\varepsilon_{kl}^0$ represents the electrostrictive stress-free strain, i.e., $\varepsilon_{kl}^0 = Q_{ijkl}P_kP_l$. $f_{elec} = f_{dipole} + f_{depola} + f_{appl}$, where $f_{dipole}$ is the dipole-dipole interactions arising from polarization, $f_{depola}$, represents the depolarization energy density, and $f_{appl}$, denoting the energy density resulting from an applied electric field. The temporal evolution of the spontaneous polarization field ($P$) can is determined by solving the time-dependent Ginzburg-Landau (TDGL) equation: $\frac{dP_i(x,t)}{dt} = -M\frac{\delta F}{\delta P_i(X,t)}$, $i = 1, 2, 3$, where $M$ represents the kinetic coefficient, $F$ signifies the total free energy, and $t$ denotes time[5,69].

## Data availability

All data supporting this study and its findings are available within the article and its Supplementary Information. The data that support the findings of this study are available on request from the corresponding authors.

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

## Acknowledgements

This work was supported by the National Key R&D Program of China (2021YFB3800602), the International Cooperation Project of Shaanxi Province (2021KWZ-10), the Fundamental Research Funds for the Central University, the 111 Project of China (B14040), Zhejiang Provincial Science and Technology Program under Grant LGG20F010007. The SEM work was done at the International Center for Dielectric Research (ICDR), Xi'an Jiaotong University, Xi'an, China. The authors thank Dr. Yan-Zhu Dai for her help with SEM.

## Author contributions

The work was conceived and designed by W.C.Z., D.Z., and W.C.Z. fabricated the samples, tested the energy storage, dielectric, structure, stability, and other properties, and processed related data, assisted by D.Z., D.X., and D.L. The SEM images were filmed and processed by G.Y. and T. Z. The HAADF-STEM images were filmed and processed by H.M.J. and D.X. The PFM images were filmed and processed by M.K.X. and W.F.L. The neutron diffraction data were processed and analyzed by D.X. and M.A. Temperature-dependent XRD data were processed and analyzed by D.E.S. The manuscript was drafted by W.C.Z. and revised by D.Z. and D.X. The phase-field simulations were performed by D.W. and discussed with D.Z. and W.C.Z. All authors participated in the data analysis and discussions.

## Competing interests

The authors declare no competing interests.
