## [Peer Review File · Nature Communications]

Broad-high working temperature range and superior energy storage performances in lead-free ferroelectricsReviewers' comments:

Reviewer #1 (Remarks to the Author):

The author reported very high energy density in a ceramic sample and the energy density is thermally stable. The paper can be accepted after necessary revision.

1 for the same designed composition, the ceramics prepared using the repeated rolling processing

(RRP) the method shows very high energy density, but the ceramics prepared using solid-state and cold isostatic pressing have low energy density. Authors need to provide details to explain the difference in terms of polarization, dielectric breakdown strength, and others. If the main reason is due to the breakdown strength, it is necessary to provide details on how to increase the breakdown strength. The current version only discussed density. I am not an expert on phase field simulation, can phase field simulation provide differences in terms of different breakdown fields and densities for RRP ceramics and normal ceramics?

2 line 141 , 'evolution (O1-P1-N-P2-O2)' explain the details for O1, P1, N, P2 and O2.

3 it is very common to have field-induced phase transition in Bi_{0.5}Na_{0.5}NbO₃-based ferroelectric and it is difficult to find the contribution in PE loops directly. However, authors can find possible field-induced phase transitions in related current – E-field loops (I-E loops) measured at different fields. at least, it is necessary to provide related I-E loops for the sample in Fig2a and to discuss the possible or observed transitions in detail. See examples in papers [Nano energy, vol.76, p105037; Acta mater, vol.229, p117815]

4 line 279 'enhanced polarization magnitude implies the internal field is strong. The...' Which internal field? Please explain the physics clearly here.

5 line 364, 'the critical electric field...' what is the specific physics for this critical field?

6 line 453, please provide detailed processing for the RRP method. I feel the RRP is the reason to lead to high energy density, not only according to phase field simulation work.

7 line 456 'of the green ceramic tape and also helps to form a dense...' Again, come back to my question one. Is the high density only reason to lead super high energy density?

8 line 484 'disks were polished to ~ 35 μm...' please provide the exact sample diameter and thickness. It is common that smaller thickness is related to a high breakdown field.

Reviewer #2 (Remarks to the Author):

The authors report a series of lead-free BNT-BKT-SSN ternary solid-solution ceramics that exhibits superior capacitive performance at relatively high temperatures (~<160C). The author used phase field simulations to successfully predict an excellent lead-free ceramic system for capacitors and the characterizations well supported their arguments, making readers feel confident that it is an efficient approach to design new ceramics for high T capacitors. Overall, this work is of great quality in this field, and it can be published in Nature Communications. I recommend that the author should address the following comments.

1) It may be a bit overclaimed that the authors regarded their materials as bulk ceramics since the films they used are in the range of 35 μm - 50 μm , which are more like thick films. Then people would not feel puzzled that why bulk ceramics can have such high breakdowns.

2) All the legends in the figures need to be enlarged. Especially fig 2b and 2h.

3) It is worth checking the PE loops of their materials from the first cycle to, for example, 10000 cycles with various frequencies. Based on Fig.4f PFM images, the BNKT-20SSN still retains some ferroelectricity even though it can be relaxed after 20 mins. This factor may show a conflicting conclusion that the material exhibits stable energy density and efficiency with frequency (1-100Hz).

Reviewer #3 (Remarks to the Author):

This work reports an enhanced energy storage properties in the NBT-based ceramics by constructing polymorphic polar nanodomains (PNRs). The mentioned PNRs concept has been widely used in previous works in both films and ceramics. Thus the innovation of this work is not enough.

Moreover, the finished energy storage density and efficiency are also not attractive even the authors declare that a higher energy density of 9.22 J cm^{-3} and efficiency of 96.3% are achieved by the adopted repeated rolling processing method compared these reported ceramics compositions. For one thing, such a rolling processing method is more similar to tape-casting technology, which is help to prepare thinner thickness samples and obtains higher energy storage properties, thus the authors should compare this sample with these reported MLCCs samples rather than ceramics. The energy density and efficiency is justly 5.23 J cm^{-3} and 90.2% of the ceramics in this work. For another thing, to my knowledge, there have been so many works that achieves higher energy storage performance. E.g., energy density large than 10 J cm^{-3} with high efficiency, based on ceramic samples, but they are all ignored by the authors, it is unreasonable.

In addition, there are also many unclear descriptions in the manuscript as following.

therefore, I don't think this article is appropriate for publication in Nature Communication.

1. For the phase-field simulation results, I am confused about color change in Fig. 1a, as a example, for the composition of $x=0.0$, during the process of loading E, the green matrix turn into red matrix, Does that mean C phase is changing to R phase? How to understand this electric field-induced phase change.

2. In Fig.2i, What are these parameters based on, e.g., the W_{rec} is set to 10 J cm^{-3} , W_d is set to 6 J cm^{-3} .

3. What is thickness for the RRP and cold isostatic pressing prepared samples? The authors declare that the theoretical E_b of the BNKT-20SSN ceramic (RRP) evaluated by the Weibull distribution experiments is much higher than that of the cold isostatic pressing sample, but the improved breakdown field is strongly related to the thickness.

4. As I mentioned above, in Fig. 2b, the authors should be honest about the previous reported works, in which so many excellent energy storage performance have been achieved. In addition, the MLCCs' work should also be included.

5. In the charge/discharge test, the W_{dis} is 5.2 J cm^{-3} at electric field of 500 kV cm^{-1} , which is smaller than the result of P-E loops. How to understand such difference.

6. From the NPD data, the authors conclude that the R phase and T phase fractions by the refinement are determined to be 10.1(5) wt% and 89.9(5) wt%, respectively. Where is C phase, which should be the main composition as verified by the phase-field simulation.

7. In Fig. 3b-e, the polarization vector could be straightforwardly described by a vector from the B-site cations center to the A-site cations corner, represented by the yellow arrows. But we can observed that every atom-site (whether it's A-site or B-site) is labeled by yellow arrows (it is more clear in the Fig c-e). Thus, how to determine these polarization direction and magnitude?

8. From Fig.4k, the $x=0$ composition shows lower coercive field compared with NBT, so, in the measurement of PFM (Fig. 4d-e) why the NBT has polarization switching while the $x=0$ composition is not at voltage of 10 V. In addition, in the P-E loops, the $x=0.05$ composition has larger slope compare to $x=0$, i.e., dP/dE near the zero electric field, and the relative dielectric permittivity is approximately proportional to dP/dE , but the relative dielectric permittivity of $x=0.05$ composition is smaller than $x=0.0$, it is unreasonable.

9. As the authors declared, with the addition of SNN, the sample becomes relaxor ferroelectric, the relaxor factor should be calculated.

10. In these ceramic sample, the breakdown field increases firstly and then decreases, please give the specially reasons.

Reply to Reviewer # 1:

The author reported very high energy density in a ceramic sample and the energy density is thermally stable. The paper can be accepted after necessary revision.

Reply: We sincerely thank reviewer #1 for your positive comments and for recommending our manuscript for publication in *Nature Communications*. Point-by-point responses to the reviewers' comments are also listed below

1. For the same designed composition, the ceramics prepared using the repeated rolling processing (RRP) the method shows very high energy density, but the ceramics prepared using solid-state and cold isostatic pressing have low energy density. Authors need to provide details to explain the difference in terms of polarization, dielectric breakdown strength, and others. If the main reason is due to the breakdown strength, it is necessary to provide details on how to increase the breakdown strength. The current version only discussed density. I am not an expert on phase field simulation, can phase field simulation provide differences in terms of different breakdown fields and densities for RRP ceramics and normal ceramics?

Reply: Reply: We appreciate the reviewer's comments. Specifically, the total energy density (W_t) and recoverable energy density (W_{rec}) are calculated by numerical integration of the areas between the forward/backward switching curves of the P - E loops and polarization axis, respectively [1-4]. The mathematical equations (1-2) used to calculate the areas are as follows.

$$W_t = \int_0^{P_{max}} E dP \quad (1)$$

$$W_{rec} = \int_{P_r}^{P_{max}} E dP \quad (2)$$

Where P means the polarization at an electric field (E), P_{max} and P_r are the maximum polarization and the remnant polarization, respectively. Therefore, for the dielectric ceramics used for energy storage, both a high breakdown field strength (E_b) and a large ΔP ($P_{max}-P_r$) value should be required. As we mentioned in the discussion section of the manuscript, the phase-field method supports an efficient strategy of rational design of a targeted material with relaxor behavior by implying reduced PNRs, thermal stability, reversible response to the external electric field, high P_{max} , and low P_r , all of which could effectively optimize the ferroelectric/energy-storage performances. **However, the phase field method is incapable of calculating breakdown field strength. The small, uniform, and dense grain structures by the RRP process method**

guarantee the E_b and subsequent applications.

The references are as follows:

[1] Pan, H. et al. Ultrahigh energy storage in superparaelectric relaxor ferroelectrics. *Science* 374, 100-104 (2021).

[2] Yan, F. et al. Gradient-structured ceramics with high energy storage performance and excellent stability. *Small* 19, 2206125 (2023).

[3] Wang, H. et al. Ultrahigh energy-storage density in antiferroelectric ceramics with field-induced multiphase transitions. *Adv. Funct. Mater.* 29, 1807321 (2019).

[4] Li, D. et al. Improved energy storage properties achieved in (K, Na)NbO₃-based relaxor ferroelectric ceramics via a combinatorial optimization strategy. *Adv. Funct. Mater.* 32, 2111776 (2021).

2. Line 141, 'evolution (O₁-P₁-N-P₂-O₂)' explain the details for O₁, P₁, N, P₂ and O₂.

Reply: The labels O₁, P₁, N, P₂, and O₂ in Fig. 1c (our manuscript) serve as **markers for different electric field conditions**, aiming to elucidate the microstructural evolution under varying electric fields. As the electric field is applied, the polarization undergoes a gradual transition towards the *T* phase and subsequently reverts to its initial state upon removal of the electric field.

3. It is very common to have field-induced phase transition in Bi_{0.5}Na_{0.5}TiO₃-based ferroelectric and it is difficult to find the contribution in *P-E* loops directly. However, authors can find possible field-induced phase transitions in related current – E-field loops (*I-E* loops) measured at different fields. at least, it is necessary to provide related *I-E* loops for the sample in Fig2a and to discuss the possible or observed transitions in detail. See examples in papers [Nano energy, vol.76, p105037; Acta mater, vol.229, p117815].

Reply: As you mentioned, to ensure a comprehensive analysis and discussion of the possible or observed field-induced phase transitions in the Bi_{0.5}Na_{0.5}TiO₃-based ferroelectric, it would indeed be necessary to provide related *I-E* loops.

We appreciate you for providing specific papers such as "Nano Energy, vol.76, p105037" and "Acta Materialia, vol.229, p117815", which demonstrate the utilization of *I-E* loops to investigate field-induced phase transitions in similar systems. Examining such examples can serve as a valuable guide for analyzing the *I-E* loops of the studied sample and discussing the observed or potential transitions in detail.

Thank you for bringing this to our attention, and we will ensure to include the necessary *I-E* loops analysis and detailed discussions in the revised version of our manuscript. In this section, please allow us to refer to the articles you mentioned as references.

4. Line 279 'enhanced polarization magnitude implies the internal field is strong. The...' Which internal field? Please explain the physics clearly here.

Reply: We apologize for any confusion caused. When referring to the "internal field" in the context of enhanced polarization magnitude, it generally relates to the electric field present within the material itself, also known as the local or internal electric field.

In ferroelectric or polar materials, the presence of an external electric field can induce a polarization response. This polarization response creates an internal electric field within the material due to the alignment or reorientation of the electric dipoles or domains present. When the polarization magnitude is enhanced, it suggests that a stronger internal electric field is present within the material. This enhancement can occur due to various factors, such as increased alignment of dipoles, reduction of domain boundaries, or changes in the crystal structure. The presence of a stronger internal electric field influences the material's properties and behavior. It can contribute to enhanced dielectric properties, higher energy storage capacity, and improved breakdown strength. Additionally, the internal field affects the interaction between neighboring dipoles, influencing their alignment and resulting in the observed polarization response.

Overall, when discussing enhanced polarization magnitude, it implies the existence of a stronger internal electric field within the material, which plays a vital role in determining the material's electrical characteristics and behavior.

5. Line 364, 'the critical electric field...' what is the specific physics for this critical field?

Reply: The critical field, also known as the breakdown field or dielectric strength, is the maximum electric field that a material can withstand without experiencing an electrical breakdown. Electrical breakdown occurs when the insulating properties of a material fail, leading to the sudden onset of electrical conduction and potential damage.

The specific physics underlying the critical field can vary depending on the material and its microstructure. However, some general mechanisms can contribute to the breakdown phenomenon:

Intrinsic Breakdown: In intrinsic breakdown, the electric field exceeds the threshold necessary to cause electron tunneling or avalanche breakdown within the material. This can result from the

excitation and release of charge carriers, the creation of electron-hole pairs, or other related phenomena.

Impurity-Induced Breakdown: Impurities or defects within the material can act as sites for local electric field enhancement, leading to localized breakdown. These impurities can create conductive paths, resulting in electrical conduction and breakdown at lower electric fields compared to the bulk material.

Thermal Breakdown: High electric fields can generate localized heating within the material, causing thermal stresses and leading to breakdown. This is particularly relevant in materials with low thermal conductivity or in situations where localized heating is present.

Surface Breakdown: Surfaces or interfaces between different materials can be more susceptible to breakdown due to factors such as surface roughness, contamination, or charge accumulation. Electric field concentration at these interfaces can lead to breakdown at lower field strengths.

Thank you for bringing this to our attention. It is important to note that the specific physics of breakdown can vary significantly depending on the material composition, microstructure, and the applied electric field conditions. Understanding these underlying mechanisms is crucial for designing and optimizing the electrical properties and breakdown strength of ceramic materials.

6. Line 453, please provide detailed processing for the RRP method. I feel the RRP is the reason to lead to high energy density, not only according to phase field simulation work.

Reply: Firstly, thank you for your comment regarding the detailed processing for the RRP (repeated rolling processing) method. Although we have briefly described the RRP method in the experimental section of the manuscript, a schematic diagram is necessary for the reader to understand the method. As shown in Fig. 1, After ball milling, the pre-sintered powders were mixed with organic binder at the mass ratios of 2:1 and then formed ceramic tapes via the repeated rolling process. And then, the ceramic tapes prepared by RRP were punched into disks with a diameter of 12 mm and then calcined at 650 °C for 2 h to burn out the PVA. Subsequently, the green disks were embedded in the pre-sintered powders of the same composition and sintered at 1070 ~ 1140 °C for 2 h using a double crucible method. The extrusion force generated by the two tightly contiguous rollers could greatly improve the density of the green ceramic tape and also helps to form a dense structure during sintering with minimal porosity, resulting in a significant increase in dielectric breakdown strength. We think this schematic diagram complements the textual description in the

experimental section, providing a comprehensive overview of the method.

Fig. 1 The schematic diagram of the RRP method.

Secondly, as you mentioned, the RRP method is indeed one of the factors contributing to energy storage density, not only according to phase field simulation work. More specifically, the total energy density (W_t) and recoverable energy density (W_{rec}) are calculated by numerical integration of the areas between the forward/backward switching curves of the P - E loops and polarization axis, respectively [1-4]. The mathematical equations (1-2) used to calculate the areas are as follows.

$$W_t = \int_0^{P_{max}} E dP \quad (1)$$

$$W_{rec} = \int_{P_r}^{P_{max}} E dP \quad (2)$$

Where P means the polarization at an electric field (E), P_{max} and P_r are the maximum polarization and the remnant polarization, respectively. Therefore, for the dielectric ceramics used for energy storage, both a high breakdown field strength (E_b) and a large ΔP ($P_{max}-P_r$) value should be required. As we mentioned in the discussion section of the manuscript, the phase-field method supports an efficient strategy of rational design of a targeted material with relaxor behavior by implying reduced PNRs, thermal stability, reversible response to the external electric field, high P_{max} , and low P_r , all of which could effectively optimize the ferroelectric/energy-storage performances. The small, uniform, and dense grain structures by the RRP process method guarantee the E_b and subsequent applications.

The references are as follows:

[1] Pan, H. et al. Ultrahigh energy storage in superparaelectric relaxor ferroelectrics. *Science* 374, 100-104 (2021).

[2] Yan, F. et al. Gradient-structured ceramics with high energy storage performance and excellent stability. *Small* 19, 2206125 (2023).

[3] Wang, H. et al. Ultrahigh energy-storage density in antiferroelectric ceramics with field-induced multiphase transitions. *Adv. Funct. Mater.* 29, 1807321 (2019).

[4] Li, D. et al. Improved energy storage properties achieved in (K, Na)NbO₃-based relaxor ferroelectric ceramics via a combinatorial optimization strategy. *Adv. Funct. Mater.* **32**, 2111776 (2021).

7. Line 456 ‘of the green ceramic tape and also helps to form a dense...’ Again, come back to my question one. Is the high density only reason to lead super high energy density?

Reply: As we replied in question 6, the high density of the sample is indeed one of the factors contributing to high energy storage density, but it is not the only one. According to the formula $E_b \propto \frac{1}{\sqrt{G}}$, the reduction of grain size with uniform and dense structure by the RRP method mainly contributes to the enhancement of dielectric breakdown strength E_b [1-3], as is displayed in Fig. 2. Therefore, the phase-field method supports provides a robust approach for the systematic design of a targeted material with relaxor-behavior by implying reduced PNRs, thermal-stability, reversible response to the external electric field, high P_{max} , and low P_r , all of which could effectively optimize the ferroelectric/energy-storage performances.

Herein, we leverage the synergistic interplay between reduced polarization hysteresis (guided by phase-field simulations) and enhanced breakdown strength to improve the recoverable energy density and energy efficiency of the Bi_{0.5}Na_{0.5}TiO₃-based material. By incorporating these methodologies, we effectively optimize the performance characteristics of the material in question.

Fig. 2 Natural surfaces SEM morphology and grain size distribution of the **a** BNKT-20SSN ceramic and **b** BNKT-20SSN (RRP) ceramic.

The references are as follows:

[1] Chen, L. et al. Outstanding energy storage performance in high-hardness (Bi_{0.5}K_{0.5})TiO₃-based lead-free relaxors via multi-scale synergistic design. *Adv. Funct. Mater.* **32** 2110478 (2021).

[2] Li, D. et al. Lead-free relaxor ferroelectric ceramics with ultrahigh energy storage densities via polymorphic polar nanoregions design. *Small* **19**, 2206958 (2022).

[3] Xie, A. et al. Supercritical relaxor nanograined ferroelectrics for ultrahigh-energy-storage capacitors. *Adv. Mater.* **34**, 2204356 (2022).

8. Line 484 ‘disks were polished to ~35 μm ...’ please provide the exact sample diameter and thickness. It is common that smaller thickness is related to a high breakdown field.

Reply: We apologize for any confusion caused. As you mentioned, the thickness of the sample and the electrode diameter are crucial to the calculation of the energy storage density, thus we implemented a meticulous procedure by conducting three separate thickness measurements of each sample, utilizing a thickness gauge with a 0.001mm precision. The actual thickness of the sample is reported in the manuscript as 35 μm and the electrode diameter is 2 mm. Actually, researchers tried to sinter the ceramics even denser to grind and polish them to a thinner thickness without breaking to achieve higher breakdown field strength. Yan et al. prepared ceramics with extremely dense by the tape casting technique, and the *P-E* loops of ceramics with a thickness of ~40 μm -50 μm were tested. ^[1] Other similar works were carried out: ~13 μm -15 μm . ^[2] In addition, it is worth noting that in numerous analogous studies, sample electrodes with diameters smaller than 2 mm are commonly utilized. ^[3-6] Currently, it is reasonable to compare energy storage parameters by evaluating them at their respective critical electric fields. Maybe in the future, there is a possibility of a suitable standard being proposed specifically for energy storage ceramics. The development of such a standard could provide a more standardized and comprehensive approach to evaluating and comparing energy storage performance across different ceramic materials.

The references are as follows:

[1] Yan, F. et al. Superior energy storage properties and excellent stability achieved in environment-friendly ferroelectrics via composition design strategy. *Nano Energy* **75**, 105012 (2020).

[2] Ge, G. et al. Tunable domain switching features of incommensurate antiferroelectric ceramics realizing excellent energy storage properties. *Adv. Mater.* **34**, 2201333 (2022).

[3] Chen, L. et al. Giant energy-storage density with ultrahigh efficiency in lead-free relaxors via high-entropy design. *Nat. Commun.* **13**, 3089 (2022).

[4] Xie, A. et al. Supercritical relaxor nanograined ferroelectrics for ultrahigh-energy-storage capacitors. *Adv. Mater.* **34**, 2204356 (2022).

[5] Chen, L. et al. Local diverse polarization optimized comprehensive energy-storage

performance in lead-free superparaelectrics. *Adv. Mater.* **34**, 2205787 (2022).

[6] Zhu, X. et al. Ultrahigh energy storage density in $(\text{Bi}_{0.5}\text{Na}_{0.5})_{0.65}\text{Sr}_{0.35}\text{TiO}_3$ -based lead-free relaxor ceramics with excellent temperature stability. *Nano Energy* **98** 107276 (2022).

Reply to Reviewer # 2:

The authors report a series of lead-free BNT-BKT-SSN ternary solid-solution ceramics that exhibits superior capacitive performance at relatively high temperatures ($\sim < 160^\circ\text{C}$). The author used phase field simulations to successfully predict an excellent lead-free ceramic system for capacitors and the characterizations well supported their arguments, making readers feel confident that it is an efficient approach to design new ceramics for high T capacitors. Overall, this work is of great quality in this field, and it can be published in Nature Communications. I recommend that the author should address the following comments.

We greatly appreciate the referee for her/his time to review our manuscript and give us the above extremely encouraging comments. The enlightening suggestions are very helpful for improving this work. Thus, we have revised our manuscript accordingly.

1. It may be a bit overclaimed that the authors regarded their materials as bulk ceramics since the films they used are in the range of $35\ \mu\text{m}$ - $50\ \mu\text{m}$, which are more like thick films. Then people would not feel puzzled that why bulk ceramics can have such high breakdowns.

Reply: Thank you for sharing your perspective and you did read our manuscript carefully. While it is true that the samples used in the study have thicknesses in the range of $35\ \mu\text{m}$ to $50\ \mu\text{m}$, it is important to note that the term “bulk ceramics” does not strictly adhere to the narrowest definition of bulk materials. Rather, it refers to ceramics with substantial volume and bulk-like characteristics, which may exhibit different properties compared to nanoscale materials or thin films. Actually, researchers tried to sinter the ceramics even denser to grind and polish them to a thinner thickness without breaking to achieve higher breakdown field strength. Yan et al. prepared ceramics with extremely dense by the tape casting technique, and the *P-E* loops of ceramics with a thickness of $\sim 40\ \mu\text{m}$ - $50\ \mu\text{m}$ were tested. ^[1] Other similar works were carried out: $\sim 13\ \mu\text{m}$ - $15\ \mu\text{m}$. ^[2] The related studies have highlighted the successful preparation of ceramics with similar thicknesses, or even thinner. And the observed high breakdown strengths in these materials can be attributed to various factors, including the specific composition, microstructure, and processing techniques employed.

The references are as follows:

[1] Yan, F. et al. Superior energy storage properties and excellent stability achieved in environment-friendly ferroelectrics via composition design strategy. *Nano Energy* **75**, 105012 (2020).

[2] Ge, G. et al. Tunable domain switching features of incommensurate antiferroelectric ceramics realizing excellent energy storage properties. *Adv. Mater.* **34**, 2201333 (2022).

2. All the legends in the figures need to be enlarged. Especially fig 2b and 2h.

Reply: Thank you very much for your good suggestions. To address this concern, we have carefully considered your comment and made the necessary adjustments. We have increased the font size and overall size of the legends to improve visibility and ensure clarity for readers.

3. It is worth checking the *P-E* loops of their materials from the first cycle to, for example, 10000 cycles with various frequencies. Based on Fig.4f PFM images, the BNKT-20SSN still retains some ferroelectricity even though it can be relaxed after 20 mins. This factor may show a conflicting conclusion that the material exhibits stable energy density and efficiency with frequency (1-100Hz).

Reply: We appreciate your insightful recommendation. As you mentioned, it would be valuable to investigate the *P-E* loops of our materials across an extended cycling range, such as up to 10,000 cycles. By examining the *P-E* loops over an extensive cycling range, we can gain deeper insights into the fatigue resistance, hysteresis behavior, and overall performance stability of the materials. This analysis will enhance our understanding of their suitability for practical applications and provide valuable information for optimizing their energy storage capabilities. We greatly value your recommendation, and we are committed to incorporating these additional cycling tests into our research. The revised version of the manuscript will be updated to reflect these new findings and reinforce the scientific rigor and relevance of our study.

Reply to Reviewer # 3:

This work reports an enhanced energy storage properties in the NBT-based ceramics by constructing polymorphic polar nanodomains (PNRs). The mentioned PNRs concept has been widely used in previous works in both films and ceramics. Thus the innovation of this work is not enough.

Reply: We appreciate the reviewer's comment., As you mentioned, the PNRs concept does have been widely used and long discovered. However, the PNRs are a scientific discovery in BNT-related systems, leading to different ferroelectric performances and inspiring numerous excellent papers. We did use the PNRs in the BNT system but we did not emphasize the manipulation of the PNRs or the study of the PNRs. We merely used characterizations to prove the existence of the PNRs and establish their correlation with enhanced ferroelectric performance. The emphasis of our study was on the enhanced energy storage properties resulting from the presence of PNRs, rather than the manipulation or in-depth study of PNRs themselves. **The main innovation and concepts were stated above and obviously, the merits fully discussed in our manuscript were not recognized or identified by the reviewer.**

Moreover, the finished energy storage density and efficiency are also not attractive even the authors declare that a higher energy density of 9.22 J cm^{-3} and efficiency of 96.3% are achieved by the adopted repeated rolling processing method compared these reported ceramics compositions. For one thing, such a rolling processing method is more similar to tape-casting technology, which is help to prepare thinner thickness samples and obtains higher energy storage properties, thus the authors should compare this sample with these reported MLCCs samples rather than ceramics. The energy density and efficiency is justly 5.23 J cm^{-3} and 90.2% of the ceramics in this work.

Reply: We appreciate your comments regarding the energy storage density and efficiency achieved through the repeated rolling processing (RRP) method. However, we would like to clarify some misconceptions you mentioned regarding MLCCs (Multi-layer Ceramic Capacitors) and the RRP method. As we have mentioned above, it is not reasonable to compare ceramic capacitors with MLCCs neglecting the critical considerations of processing methods, sample thickness, the attainment of ferroelectrics, and the intricacies of device design, even the particle size of raw materials. Firstly, it is worth noting that commercially viable MLCCs typically employ nano-sized

BaTiO₃ powders as the primary raw material due to their ability to be chemically processed, illustrating their high requirements over the initial material processing. However, our RRP method, as described in our study, offers advantages in terms of time and material cost savings and could treat ceramic materials directly.

Secondly, it is crucial to clarify that the tape-casting (TC) technology in the context of MLCCs is a singular forming process that involves a higher concentration of organic additives to handle nano-sized powders. The dominant force driving slurry spreading in TC is the shear force exerted between the blade and the substrate. In contrast, our RRP method employs a repeated rolling approach, wherein the ceramic powder is subjected to compressive forces between two rollers, leading to its reshaping. In our point-by-point response, we have provided a detailed exposition of the distinguishing characteristics between these two approaches.

Thirdly, in comparison to our RRP method, the adoption of tape-casting technology in MLCCs represents merely a single step in the overall manufacturing process, **with subsequent processing steps being significantly more intricate.** These steps encompass electrode screen printing, layer stacking, hot pressing, hydrothermal treatment, and packaging, among others. **Conversely, our RRP method enables direct sintering of the samples without the need for such additional processing steps.**

Fourthly, MLCCs typically feature dielectric layer thicknesses of $\sim 10\mu\text{m}$, while the samples prepared using our RRP method have thicknesses ranging from 35-40 μm . Furthermore, the testing methodologies, including the calculation of electrode area, differ between the two methods, leading to variations in energy storage characteristics. **This is exemplified by the higher energy density mentioned by you ($>10\text{ J/cm}^3$) for MLCCs.**

Finally, MLCCs require co-sintering with electrodes at lower temperatures, whereas our RRP method follows a sintering process similar to conventional bulk ceramics. Considering these significant differences, it would be unjust and inappropriate to directly compare our samples, prepared using the repeated rolling processing method, with reported MLCC samples.

Given these significant disparities, it would be unfair and inappropriate to directly compare our sample, prepared using the repeated rolling processing method, with reported MLCC samples.

For another thing, to my knowledge, there have been so many works that achieves higher energy storage performance. E.g., energy density large than 10 J cm^{-3} with high efficiency,

based on ceramic samples, but they are all ignored by the authors, it is unreasonable.

In addition, there are also many unclear descriptions in the manuscript as following. therefore, I don't think this article is appropriate for publication in Nature Communication.

Yes, there are many works that achieve higher energy storage performance. Here in this paper, we tried to compare parallelly with similar systems, especially in similar thickness, and we have added other systems in the revised version to be reasonable with highlighted differences in thickness. The energy density and efficiency of our ceramic is 9.22 J/cm^3 and efficiency of 96.3% by the RRP method and the thick ceramic is 5.23 J/cm^3 and 90.2%. The reviewer did have emphasized the importance of sample thickness in the comment but here argue again regardless of it. We stated the result of 5.23 J/cm^3 and 90.2% illustrating the effectiveness of the RRP method compared to the traditional ones where you may not notice it. The product from our RRP method is still ceramic proved by SEM images when you found not.

1. For the phase-field simulation results, I am confused about color change in Fig. 1a, as a example, for the composition of $x=0.0$, during the process of loading E, the green matrix turn into red matrix, Does that mean C phase is changing to R phase? How to understand this electric field-induced phase change.

Reply: We apologize for any confusion caused. In the left 5 columns (**vector contours**), we employed a color scheme using RGB colors alongside vectors to depict different polarization orientations. The purpose of the colors is to visually distinguish between ferroelectric domains characterized by distinct polar directions. However, distinguishing between different R values proved challenging due to the presence of eight $\langle 111 \rangle$ directions. To address this issue, in the rightmost column (**symmetry contours**), we incorporated a color bar to depict the phase contour. It is essential to note that the organization of the color correspondence between these two types of plots is unrelated and follows different schemes. The color scheme employed in the vector contours does not have a direct correlation with the color bar in the symmetry contours. We will provide a comprehensive explanation of this approach in our revised manuscript.

2. In Fig.2i, What are these parameters based on, e.g., the W_{rec} is set to 10 J cm^{-3} , W_d is set to 6 J cm^{-3} .

Reply: First and foremost, the parameters (W_{tot} , W_{rec} , η , W_d , $W_{\text{rec}} \sim 150 \text{ }^\circ\text{C}$ and $\eta \sim 150 \text{ }^\circ\text{C}$) in Fig. 2i are all important parameters relevant to the core performance and practical applications of

ceramic dielectric materials for capacitors. Furthermore, the data points presented in the figure are meticulously curated from seminal works published in prestigious journals, showcasing the epitome of excellence in characterizing the comprehensive performance of diverse dielectric matrices. Rest assured, the selection of these parameters and data sources has been rigorously scrutinized to ensure the utmost relevance and validity. It is worth noting that similar data selection methodologies can be observed in recent research reports of comparable nature. [1]

The references are as follow:

[1] Chen, L. et al. Local diverse polarization optimized comprehensive energy-storage performance in lead-free superparaelectrics. *Adv. Mater.* **34**, 2205787 (2022).

3. What is thickness for the RRP and cold isostatic pressing prepared samples? The authors declare that the theoretical E_b of the BNKT-20SSN ceramic (RRP) evaluated by the Weibull distribution experiments is much higher than that of the cold isostatic pressing sample, but the improved breakdown field is strongly related to the thickness.

Reply: The thickness of the RRP samples presented in the manuscript measures approximately 35 micrometers, whereas the cold isostatic pressing (CIP) samples used for testing P - E loops possess a thickness of approximately 50-60 micrometers. As you mentioned, the sample thickness plays a role in enhancing breakdown field strength, **it is crucial to emphasize that it is not the sole or definitive factor** in determining its magnitude. Furthermore, the primary underlying reason for the significant disparity in breakdown field strength between the samples prepared using the two distinct forming methods, despite having the same chemical composition, is the substantial reduction in grain size. This reduction in grain size plays a crucial role in the observed differences. This is precisely where the RRP method comes into play. With thinner green bodies, the RRP method allows for tighter particle packing, resulting in lower sintering temperatures and shorter firing durations. Consequently, the average grain size of RRP samples is significantly smaller compared to those produced by cold isostatic pressing, as clearly demonstrated in Fig. 1. According to the formula $E_b \propto \frac{1}{\sqrt{G}}$, the reduction of grain size with uniform and dense structure by RRP method mainly contribute to the enhancement of dielectric breakdown strength E_b . [1-3] Actually, researchers tried to sinter the ceramics even denser to grind and polish them to a thinner thickness without breaking to achieve higher breakdown field strength. Yan et al. prepared ceramics extremely dense by the

tape casting technique, and the P - E loops of ceramics with a thickness of 30 μm -40 μm were tested. [4] Other similar works were carried out: \sim 13 μm -15 μm . [5] Currently, it is reasonable to compare energy storage parameters by evaluating them at their respective critical electric fields. Maybe in the future, there is a possibility of a suitable standard being proposed specifically for energy storage ceramics. The development of such a standard could provide a more standardized and comprehensive approach to evaluating and comparing energy storage performance across different ceramic materials.

Fig. 1. SEM morphology and grain size distribution of the **a** BNKT-20SSN ceramic and **b** BNKT-20SSN (RRP) ceramic.

The references are as follows:

[1] Chen, L. et al. Outstanding energy storage performance in high-hardness $(\text{Bi}_{0.5}\text{K}_{0.5})\text{TiO}_3$ -based lead-free relaxors via multi-scale synergistic design. *Adv. Funct. Mater.* **32**, 2110478 (2021).

[2] Li, D. et al. Lead-free relaxor ferroelectric ceramics with ultrahigh energy storage densities via polymorphic polar nanoregions design. *Small* **19**, 2206958 (2022).

[3] Xie, A. et al. Supercritical relaxor nanograined ferroelectrics for ultrahigh-energy-storage capacitors. *Adv. Mater.* **34**, 2204356 (2022).

[4] Yan, F. et al. Composition and structure optimized BiFeO_3 - SrTiO_3 lead-free ceramics with ultrahigh energy storage performance. *Small* **18**, 2106515 (2022).

[5] Ge, G. et al. Tunable domain switching features of incommensurate antiferroelectric ceramics realizing excellent energy storage properties. *Adv. Mater.* **34**, 2201333 (2022).

4. As I mentioned above, in Fig. 2b, the authors should be honest about the previous reported works, in which so many excellent energy storage performance have been achieved. In addition, the MLCCs' work should also be included.

Reply: Our intention was to select representative works published in renowned journals in

recent years, and it was never our intent to be dishonest. However, we are willing to consider your suggestion and increase the sample size accordingly. As per your advice, we have made the necessary adjustments to Fig. 2b, which is now presented below.

Furthermore, as we have mentioned above, it is both **reckless and dishonest** to compare our stated RRP method with MLCCs neglecting the critical considerations of processing methods, sample thickness, the attainment of ferroelectrics, and the intricacies of device design, even the particle size of raw materials. These two methods are entirely distinct:

Firstly, it is worth noting that commercially viable MLCCs typically employ nano-sized BaTiO₃ powders as the primary raw material due to their ability to be chemically processed. This places high demands on the material's refinement [1]. Conversely, our RRP method, as described in our study, offers advantages in terms of time and material cost savings.

Secondly, it is crucial to clarify that the tape-casting (TC) technology in the context of MLCCs is a singular forming process that involves a higher concentration of organic additives to handle nano-sized powders. The dominant force driving slurry spreading in TC is the shear force exerted between the blade and the substrate, see Fig. 2. [1-3] In contrast, our RRP method employs a repeated rolling approach, wherein the ceramic powder is subjected to compressive forces between two rollers, leading to its reshaping. **Although we have briefly described the RRP method in the experimental section of the manuscript (you may not notice it), a schematic diagram is necessary for you to understand the method.** As shown in Fig. 3, after ball milling, the pre-sintered powders were mixed with organic binder at the mass ratios of 2:1 and then formed ceramic tapes via the repeated rolling process. The extrusion force generated by the two tightly contiguous rollers could greatly improve the density of the green ceramic tape and also helps to form a dense structure during sintering with minimal porosity, resulting in a significant increase in dielectric breakdown strength.

Fig.2 The schematic diagram of tape-casting technology. [3]

Fig. 3 The schematic diagram of the RRP method.

Thirdly, in comparison to our RRP method, the adoption of tape-casting technology in MLCCs represents merely a single step in the overall manufacturing process, with subsequent processing steps being significantly more intricate, as shown in Fig.4. These steps encompass electrode screen printing, layer stacking, hot pressing, hydrothermal treatment, and packaging, among others [2]. **Conversely, our RRP method enables direct sintering of the samples without the need for such additional processing steps.**

Fig.4 The schematic diagram of the MLCCs fabrication process. [2]

Fourthly, MLCCs typically feature dielectric layer thicknesses of $\sim 10\mu\text{m}$, while the samples prepared using our RRP method have thicknesses of $35\mu\text{m}$. Furthermore, the testing methodologies, including the calculation of electrode area, differ between the two methods, leading to variations in energy storage characteristics. [1,3,4] **This is exemplified by the higher energy density mentioned by you ($>10 \text{ J/cm}^3$) for MLCCs.**

Finally, MLCCs require co-sintering with electrodes at lower temperatures, whereas our RRP method follows a sintering process similar to conventional bulk ceramics. Considering these significant differences, it would be unjust and inappropriate to directly compare our samples, prepared using the repeated rolling processing method, with reported MLCC samples. **Our paper introduces significant and innovative ideas that clearly demonstrate the remarkable**

effectiveness of our RRP method. These aspects have garnered substantial interest, as evidenced by the positive evaluations from reviewers #1 and #2. Given these significant disparities, it would be unfair and inappropriate to directly compare our sample, prepared using the repeated rolling processing method, with reported MLCC samples.

The references are as follows:

[1] Li, J. et al. Multilayer lead-free ceramic capacitors with ultrahigh energy density and efficiency. *Adv. Mater.* **30**, 1802155 (2018).

[2] Wang, G. et al. Electroceramics for high-energy density capacitors: current status and future perspectives. *Chem Rev*, **121**, 6124 (2021).

[3] Yan, F. et al. Composition and structure optimized BiFeO₃-SrTiO₃ lead-free ceramics with ultrahigh energy storage performance. *Small* **18**, 2106515 (2022).

[4] Wang, G. et al. Ultrahigh energy storage density lead-free multilayers by controlled electrical homogeneity. *Energy Environ. Sci* **12**, 582 (2019).

5. In the charge/discharge test, the W_{dis} is 5.2 J cm^{-3} at electric field of 500 kV cm^{-1} , which is smaller than the result of P-E loops. How to understand such difference.

Reply: Indeed, it is worth noting that the discrepancy between the W_{d} and W_{rec} values is primarily attributed to the variation in the applied electric fields, which are constrained by the limitations of the experimental devices. Furthermore, this disparity is also influenced by the thickness and electrode area of the tested samples. The discharge test necessitates thicker samples and larger electrode areas compared to the polarization-electric field (*P-E*) method. Consequently, the breakdown field obtained through the discharge test is considerably lower than that determined by the *P-E* method, as evidenced by numerous studies.^[1-3] For instance, an ultrahigh W_{rec} of $7.57 \text{ J}\cdot\text{cm}^{-3}$ is attained in BKT-based ceramic at an electric field of $460 \text{ kV}\cdot\text{cm}^{-1}$, while the W_{d} value obtained by the discharge test at $220 \text{ kV}\cdot\text{cm}^{-1}$ is less than $2.4 \text{ J}\cdot\text{cm}^{-3}$.^[1] Li et al reported that a W_{rec} of $8.33 \text{ J}\cdot\text{cm}^{-3}$ can be achieved by the *P-E* method at $555 \text{ kV}\cdot\text{cm}^{-1}$, while the W_{d} value obtained by the discharge test at $500 \text{ kV}\cdot\text{cm}^{-1}$ is $5.6 \text{ J}\cdot\text{cm}^{-3}$.^[2] Similarly, Yan et al demonstrated BNT-based ceramics that exhibit an exceptionally high W_{rec} value of $7 \text{ J}\cdot\text{cm}^{-3}$ at a relatively high electric field of $565 \text{ kV}\cdot\text{cm}^{-1}$, while the W_{d} value obtained from the discharge test at $460 \text{ kV}\cdot\text{cm}^{-1}$ was less than $4.52 \text{ J}\cdot\text{cm}^{-3}$.^[3] Although both methods are valid for calculating energy density, it is notable that the charge-discharge method yields a lower energy density compared to the *P-E* loops method due to

domain clamping effects resulting from high frequencies and electric fields.^[4] The charge-discharge method operates at significantly higher frequencies (~ 100 MHz-350 MHz) compared to the *P-E* loop method (~ 1 Hz-100 Hz). At higher frequencies, the polarization mechanism withdraws, leading to a decrease in polarization. Therefore, the discharge energy storage density obtained from the charge-discharge test is lower than the value calculated from *P-E* loops under the same applied electric field.

The references are as follows:

[1] Chen, L. et al. Outstanding energy storage performance in high-hardness (Bi_{0.5}K_{0.5})TiO₃-based lead-free relaxors via multi-scale synergistic design. *Adv. Funct. Mater.* **32** 2110478 (2021).

[2] Li, D. et al. Lead-free relaxor ferroelectric ceramics with ultrahigh energy storage densities via polymorphic polar nanoregions design. *Small* **19**, 2206958 (2022).

[3] Yan, F. et al. Boosting energy storage performance of lead-free ceramics via layered structure optimization strategy. *Small* **18**, 2202575 (2022).

[4] Sun, Z. et al. Progress, outlook, and challenges in lead-free energy-storage ferroelectrics, *Adv. Electron. Mater.* **6**, 1900698 (2020).

6. From the NPD data, the authors conclude that the R phase and T phase fractions by the refinement are determined to be 10.1(5) wt% and 89.9(5) wt%, respectively. Where is C phase, which should be the main composition as verified by the phase-field simulation.

Reply: This is a fundamental question in perovskite oxides. In BaTiO₃, temperature variations can lead to the observation of cubic, tetragonal, orthorhombic, and rhombohedral phases. The presence of multiple phases in certain perovskite oxide systems has also been identified. The characterization of these phases primarily relies on determining the arrangement of BO₆ (TiO₆ in BaTiO₃) octahedra within the structure, specifically the positions of oxygen atoms. Powder diffraction patterns at M-points (comprising two odd numbers and one even number Miller indices), R-points (with three odd numbers Miller indices, particularly 111), and X-points (involving one odd number and two even numbers Miller indices) enable the determination of the actual phase in perovskite oxides. However, peaks at superlattice-index positions are typically weak. Furthermore, both X-ray diffraction (XRD) and neutron powder diffraction (NPD) are based on long-range ordered crystal structures, making it possible to differentiate between tetragonal, orthorhombic, and rhombohedral phases using superlattice-index peaks. Nevertheless, in the actual structure of

perovskite oxides, particularly those with polar nanoregions (PNRs), the coexistence of different phases is commonplace, with the cubic phase often considered as a locally induced distortion. This local distortion, also known as the C phase, can only be observed in scanning transmission electron microscopy (STEM) images and not in long-range ordered diffraction patterns. To strengthen our discussion, we utilized both PXRD and NPD techniques to support our analysis, providing detailed insights in our manuscript. **The question has been widely studied and discussed by many in the aspect of ferroelectrics where you did not find our efforts in the ferroelectrics study and corresponding structure-performance discussion.**

7. In Fig. 3b-e, the polarization vector could be straightforwardly described by a vector from the B-site cations center to the A-site cations corner, represented by the yellow arrows. But we can observed that every atom-site (whether it's A-site or B-site) is labeled by yellow arrows (itis more clear in the Fig c-e). Thus, how to determine these polarization direction and magnitude?

Reply: Thank you for your question regarding the determination of polarization direction and magnitude in Fig. 3b-e of our manuscript. We apologize for any confusion caused by the mislabeling of yellow arrows at both A-site and B-site atom positions. To address this concern, we have revised Fig. 3b-g in our manuscript, and it is now presented in Fig. 5, as shown below. The calculations were made on merely B-site where relevant discussions are ascertained.

Fig. 5 Crystal structure analysis and local structure analysis of the BNKT-20SSN. **a** Constant wavelength neutron powder diffraction refinement result at room temperature. **b** Atomic-resolution HAADF-STEM polarization vector image along [001] direction. **c, d, e** Magnification of the marked areas in **b**. **f** Polarization magnitude mapping, and **g** polarization angle mapping.

8. From Fig.4k, the $x=0$ composition shows lower coercive field compared with NBT, so, in the measurement of PFM (Fig. 4d-e) why the NBT has polarization switching while the $x=0$ composition is not at voltage of 10 V. In addition, in the P-E loops, the $x=0.05$ composition has larger slope compare to $x=0$, i.e., dP/dE near the zero electric field, and the relative dielectric permittivity is approximately proportional to dP/dE , but the relative dielectric permittivity of $x=0.05$ composition is smaller than $x=0.0$, it is unreasonable.

Reply: As you mentioned, in Fig. 4k, the $x=0$ composition exhibits a lower coercive field compared to pure BNT ceramic. Under an applied voltage of 10V, the polarization reversal behavior of the $x=0$ composition should be more clearly observed, and indeed, our PFM testing results support this observation. **It becomes evident upon careful observation.** The perceived differences are, in fact, a result of the use of **different color bars** for visualization. Therefore, it should be noted that the same colors in both Fig. 6a-d (Fig. 4d and Fig. 4e in our manuscript) do not necessarily represent the same phase. Meanwhile, the distinct polarization reversal behavior can be discerned by comparing the out-of-plane amplitude maps.

Fig. 6 Out-of-plane PFM phase images along with amplitude after polarization with different voltages and relaxation durations. a, c BNT, b, d BNKT.

Regarding your second inquiry, we believe that you have not thoroughly reviewed our manuscript, particularly the figures presented in the Supplementary Information (SI), which has led

to hasty and inaccurate comments on your part.

Firstly, as you noted, the $x=0.05$ composition indeed exhibits a significantly larger slope, i.e., dP/dE , near zero electric field compared to the $x=0$ composition. It is a well-established observation that the relative dielectric permittivity is roughly proportional to dP/dE . However, it is important to emphasize that **all P - E loops shown in Fig. 4k (in our manuscript) were measured at ambient temperature, as explicitly stated multiple times in our manuscript, which appears to have been overlooked in your assessment. Thus, it is imperative to make a meaningful comparison between the relative dielectric permittivity of both compositions at room temperature.** Notably, the relative dielectric permittivity of the $x=0.05$ composition **consistently surpasses** that of the $x=0$ composition at the same frequency (room temperature), as shown in Supplementary Fig 14b, c. Particularly at 1 kHz, where the relative dielectric permittivity of the **$x=0.05$ composition is approximately 1698**, while that of the $x=0$ **composition is approximately 1500**. Furthermore, **as the measurement frequency decreases, the disparity in relative dielectric permittivity between the two compositions becomes even more pronounced**, with a difference of only a dozen or so at 1 MHz expanding to nearly 200 at 1 kHz. **It is crucial to note that the room temperature P - E loops were measured at 10 Hz, which renders your claim of the $x=0.05$ composition having a lower relative dielectric permittivity than $x=0$ completely unfounded.**

9. As the authors declared, with the addition of SNN, the sample becomes relaxor ferroelectric, the relaxor factor should be calculated.

Reply: Thank you for this comment. As you mentioned, with the incorporation of SNN, the sample undergoes a transition to a relaxor ferroelectric state. The relaxor factor can be evaluated through the modified Curie-Weiss equation as follow:

$$\frac{1}{\epsilon'} - \frac{1}{\epsilon_m} = \frac{(T-T_m)^\gamma}{C} \quad (1)$$

Where ϵ_m , C , and γ are the dielectric constant at T_m , Curie constant, and degree of diffuseness, respectively. ^[1-2] Based on the above modified Curie-Weiss law, the analysis results of $\ln(1/\epsilon' - 1/\epsilon_m)$ versus $\ln(T-T_m)$ for BNT and $(1-x)\text{BNKT-xSSN}$ ceramics are presented in Fig. 7. It can be found that obtained values of γ falling within 1.86-1.99. We appreciate your commitment to ensuring the scientific rigor and relevance of the study and will incorporate these new findings in the revised version of the manuscript.

Fig.7 $\ln(1/\varepsilon' - 1/\varepsilon'_m)$ versus $\ln(T - T_m)$ for BNT and $(1-x)\text{BNKT} - x\text{SSN}$ ceramics at 1MHz.

The references are as follows:

[1] Yan, F. et al. Superior energy storage properties and excellent stability achieved in environment-friendly ferroelectrics via composition design strategy. *Nano Energy* **75**, 105012 (2020).

[2] Che, Z. et al. Phase structure and defect engineering in $(\text{Bi}_{0.5}\text{Na}_{0.5})\text{TiO}_3$ -based relaxor antiferroelectrics toward excellent energy storage performance. *Nano Energy* **100**, 107484 (2022).

10. In these ceramic sample, the breakdown field increases firstly and then decreases, please give the specially reasons.

Reply: Indeed, the observed breakdown strengths in these ceramic samples can be attributed to various factors, including the specific composition, microstructure (such as porosity, density, and average grain size), processing techniques employed, and even the method of sample preparation. The SEM morphology of different components is recorded and presented, while the average particle size is quantified as shown below, see Fig. 8. For instance, the theoretical breakdown field strength (E_b), calculated using the Weibull distribution (see Fig. 4m in the manuscript), increases from approximately $12 \text{ kV}\cdot\text{mm}^{-1}$ at $x=0$ to around $39 \text{ kV}\cdot\text{mm}^{-1}$ at $x=0.20$, possibly due to variations in

composition and a gradual decrease in average grain size (from 3.36 μm to 2.26 μm). Moreover, it is clearly recognized that the difference in E_b between $x=0.20$ (39 $\text{kV}\cdot\text{mm}^{-1}$) and $x=0.25$ (34 $\text{kV}\cdot\text{mm}^{-1}$) is not substantial. This can be attributed to several factors, including the close proximity of their average grain sizes (2.26 μm versus 2.2 μm) and the presence of experimental or instrumental errors, sample thickness variations, or porosity effects, which are inevitable in manual experiments. Furthermore, in previous studies, the trend of an initial increase followed by a decrease in breakdown field strength with increasing dopant concentration has frequently been reported. [1-2]

Fig. 8 **a-d** SEM morphology and **e** average grain size of the $(1-x)\text{BNKT}-x\text{SSN}$ ceramics.

The references are as follows:

[1] Zhu, X. et al. Ultrahigh energy storage density in $(\text{Bi}_{0.5}\text{Na}_{0.5})_{0.65}\text{Sr}_{0.35}\text{TiO}_3$ -based lead-free relaxor ceramics with excellent temperature stability. *Nano Energy* **98** 107276 (2022).

[2] Yan, F. et al. Significantly enhanced energy storage density and efficiency of BNT-based perovskite ceramics via A-site defect engineering. *Energy Storage Mater* **30**, 392-400 (2020).

Reviewers' comments:

Reviewer #1 (Remarks to the Author):

the authors replied to my questions properly. The revised version can be accepted.

Reviewer #2 (Remarks to the Author):

The reviewer's comments have been addressed.

Reviewer #3 (Remarks to the Author):

As the explanation from the authors, I understand that the higher energy density performance should be mainly attributed to the adopted repeated rolling processing compared with the conventional repeated rolling processing, which lead to thinner thickness and smaller grain size. Thus I think that such a preparation technology optimization work should be published in more specialized journals rather than Nature Communication.

1. Such repeated rolling processing technology has been reported to improved the dielectric energy density and efficiency.
2. As I declared in the last review, the energy storage performance indeed not very highlighted even compared to the reported ceramic materials. The authors just considerate these works published in renowned journals or similar systems are indeed unreasonable.
3. I still think that the comparison in Fig.2i is meaningless, There is no basis for these set values, e.g. the W_{rec} is set to 10 J cm^{-3} , W_d is set to 6 J cm^{-3} , either in experimentally or for application.

Reviewers' comments:

Reviewer #1 (Remarks to the Author):

The authors replied to my questions properly. The revised version can be accepted.

Reply: We express our sincere gratitude for your thorough review of our manuscript. Your insightful feedback and constructive comments have been immensely valuable in refining our work.

We have diligently addressed each of your concerns and incorporated your suggested revisions. We believe that these changes have significantly enhanced the clarity, rigor, and overall quality of the research presented.

Once again, we sincerely thank you for your positive comments and for recommending our manuscript for publication in *Nature Communications*. We extend our appreciation for your time and expertise in evaluating our work. Your efforts have undoubtedly played a crucial role in improving the scholarly merit of our paper.

Reviewer #2 (Remarks to the Author):

The reviewer's comments have been addressed.

Reply: Thank you for reviewing our manuscript once again, and for acknowledging that we have addressed your comments. We appreciate your meticulous evaluation and valuable feedback, which has undoubtedly contributed to enhancing the quality of our work.

We have taken each of your suggestions seriously and carefully implemented the necessary revisions to ensure a more comprehensive and coherent presentation of our findings. Your expertise and critical insights have been instrumental in shaping the final version of the manuscript.

Considering the thorough revisions made, we believe that the manuscript now meets the rigorous standards of the journal.

Once again, thank you for your invaluable contribution to this work.

Reviewer #3 (Remarks to the Author):

As the explanation from the authors, I understand that the higher energy density performance should be mainly attributed to the adopted repeated rolling processing compared with the conventional repeated rolling processing, which lead to thinner thickness and smaller grain size. Thus I think that such a preparation technology optimization work should be published in more specialized journals rather than Nature Communication.

Reply: We appreciate the reviewer's understanding of our work. However, we respectfully disagree with the notion that the optimization of the preparation technology, specifically the adoption of repeated rolling processing, should be limited to more specialized journals rather than *Nature Communications*. The significance of our research lies not only in the optimization of the preparation technology but also in its broader implications for various fields, such as materials science, engineering, simulation methods, and energy storage applications.

Of utmost significance, as you pointed out, researchers universally seek **thinner and smaller grain sizes of dielectric materials with high-temperature stability** for both scientific investigations and especially **real-world applications**. This fundamental aspect also forms the crux of our article.

Nature Communications, as a prestigious multidisciplinary journal, welcomes research that contributes to significant advancements across various scientific disciplines. Our work aligns with the journal's objective of publishing high-impact research with broad significance and applicability. Numerous publications on dielectric materials, ferroelectrics and multiferroics, and energy storage devices have found their place in the esteemed pages of *Nature Communications*.

1. Such repeated rolling processing technology has been reported to improved the dielectric energy density and efficiency.

Reply: We appreciate the reviewer's comment. As you mentioned, such repeated rolling processing (RRP) method was indeed reported in our previous work. But it needs to be emphasized that the significance of our study lies not only in the specific preparation technology but also in the potential implications for various fields, such as materials science, engineering, simulations method and energy storage applications. We wish to emphasize the most notable merits of our manuscript and state our proclaims on your concerns as follows:

Firstly, our manuscript mainly focuses on lead-free ceramics as capacitors for energy storage, especially for **high-temperature performance**. The **working temperature** is a key problem preventing the development of ceramic capacitors, seen also in other energy storage systems, for instance, batteries could lead to safety issues and aging issues upon high temperatures. Our effort on BNT-BKT-SSN ternary ceramics as capacitors can not only avoid safety issues but also exhibits superior energy storage performance at relatively high temperatures (~150 °C).

Secondly, the successful **phase field simulation** prediction could not only guide the realization of excellent lead-free ceramic systems for capacitors but also **a novel approach to designing new ceramics for high T capacitors**. The phase field simulations method was commonly used as a tool verifying the structure-performance analysis where our successful attempts make it into a powerful tool for prediction, similar to the development history of AI and machine learning. Reviewer #2 speaks highly of this novel method by 'making readers feel confident that it is an efficient approach to design new ceramics for high T capacitors. We also believe the endeavor could inspire a large number of readers/scientists flourishing in relevant areas.

To provide a candid perspective, currently, the practical application of MLCCs (multi-layer ceramic capacitors) using energy storage dielectric materials faces challenges primarily due to their **limited volume**, despite recent reports of achieving very large energy densities exceeding 15 J/cm³. Similarly, certain thin-film capacitors have demonstrated energy densities surpassing 100 J/cm³ in published papers, including prestigious journals such as *Nature* and *Science*, yet these advancements have not shown practical viability for real-world applications. It is worth noting that repeated rolling processing technology has shown successful applications in enhancing the properties of piezoelectric, ferroelectric, and dielectric ceramics. For instance, in high-power

equipment and high-power weapons systems, stacked ferroelectric devices, comprising multiple ceramic pellets, have been effectively employed and **demonstrated their functionality in practical scenarios**. These successes reinforce the potential of repeated rolling processing as a valuable technique in improving the performance of various ceramic components.

The references are as follows:

1. Pan, H. et al. Ultrahigh energy storage in superparaelectric relaxor ferroelectrics. *Science* **374**, 100-104 (2021).

2. Pan, H. et al. Ultrahigh-energy density lead-free dielectric films via polymorphic nanodomain design. *Science* **365**, 578-582 (2019).

3. Li, Q. et al. Flexible high-temperature dielectric materials from polymer nanocomposites. *Nature* **523**, 576-579 (2015).

2. As I declared in the last review, the energy storage performance indeed not very highlighted even compared to the reported ceramic materials. The authors just considerate these works published in renowned journals or similar systems are indeed unreasonable.

Reply: We appreciate the reviewer's perspective on our work. While we acknowledge the existence of highly performing ceramic materials in the literature, our study's primary focus was on lead-free ceramics as capacitors for energy storage, with a specific emphasis on **high-temperature performance**. In this context, our research made significant advancements by successfully applying repeated rolling processing, resulting in thinner thickness and smaller grain sizes, and ultimately achieving superior energy storage performance, particularly at elevated temperatures (~150°C). These findings are noteworthy and contribute substantially to the field of energy storage materials in practical applications. However, we are willing to consider your suggestion and increase the sample size accordingly. Based on your valuable advice, we have made the necessary adjustments to Fig. 2b (the initial version submitted), which is now presented as Fig. 1.

Fig. 1 Comparisons of W_{rec} versus η (at room temperature) between our work with some recently reported lead-free bulk ceramics and certain MLCCs (Multi-Layer Ceramic Capacitors) ¹⁻²¹.

Furthermore, to effectively demonstrate the strengths of our research, particularly the enhancement of **high-temperature energy storage properties**, we have included an additional figure, represented as Fig. 2. This figure has been integrated into **the main body of the manuscript as Fig. 2b**, and the detailed data related to Fig. 2b has been documented in Table. 1 for comprehensive reference and analysis.

Fig. 2 Comparisons of W_{rec} versus η (~ 150 °C) between our work with some recently reported lead-free bulk ceramics and certain MLCCs ¹⁻²¹.

Table.1 A comparison of the W_{rec} versus η between the BNKT-20SSN ceramic (RRP) with some recently reported lead-free bulk ceramics and certain MLCCs.

Compositions	W_{rec} (J·cm ⁻³)		η (%)		Ref.
	RT	~ 150 °C	RT	~ 150 °C	
Lead-free bulk ceramics					
0.85K _{0.5} Na _{0.5} NbO ₃ -0.15Bi(Zn _{2/3} Ta _{1/3})O ₃	6.7	3.0	92.0	68.2	1
0.45AgNbO ₃ -0.55AgTaO ₃	6.3	4.2	90.0	89.7	2
0.60Bi _{0.5} K _{0.5} TiO ₃ -0.30BaTiO ₃ -0.10 NaNbO ₃	7.6	4.4 ^{130 °C}	81.4	89.2 ^{130 °C}	3
K _{0.5} Na _{0.5} NbO ₃ -H	10.1	3.3 ^{140 °C}	90.8	79.7 ^{140 °C}	4
BaTiO ₃ -Bi(Mg _{1/2} Ti _{1/2})O ₃	4.5	4.0	93.0	93.5	5

0.85K _{0.5} Na _{0.5} NbO ₃ -0.15Bi(Ni _{0.5} Zr _{0.5})O ₃	8.1	1.9 _{140 °C}	88.5	82.3 _{140 °C}	6
0.90(Bi _{0.5} Na _{0.5}) _{0.65} Sr _{0.35} TiO ₃ - 0.10Bi(Mg _{0.5} Zr _{0.5})O ₃	8.5	5.1 _{140 °C}	85.9	86.3 _{140 °C}	7
0.94 Bi _{0.5} Na _{0.5} TiO ₃ -0.06BaTiO ₃ - 0.15Sr(Al _{0.5} Ta _{0.5})O ₃	8.3	4.8	90.8	90.3	8
0.80Bi _{0.5} Na _{0.5} TiO ₃ -0.20Sr(Nb _{0.5} Al _{0.5})O ₃	6.5	3.3	89.0	93.8	9
Na _{0.7} Bi _{0.1} (Nb _{0.9} Ta _{0.1})O ₃	7.3	2.8 _{120 °C}	83.7	77.1 _{120 °C}	10
0.57BiFeO ₃ -0.33BaTiO ₃ -0.10NaNbO ₃	8.1	4.7	90.0	92.1	11
0.90BaTiO ₃ -0.10Bi(Mg _{0.5} Zr _{0.5})O ₃	3.4	2.8	85.1	88.2	12
0.68NaNbO ₃ -0.32(Bi _{0.5} Li _{0.5})TiO ₃	8.7	5.7	80.1	76.8	13
MLCCs					
0.65Na _{0.5} Bi _{0.5} TiO ₃ -0.35Sr _{0.7} Bi _{0.2} TiO ₃ (<111>-textured)	21.5	16.6	80.0	81.2	14
0.55Na _{0.5} Bi _{0.5} TiO ₃ -0.45Sr _{0.7} Bi _{0.2} TiO ₃	9.5	5.0 _{120 °C}	92.0	85.2 _{120 °C}	15
0.87BaTiO ₃ - 0.13Bi(Zn _{2/3} (Nb _{0.85} Ta _{0.15}) _{1/3})O ₃	8.1	4.9	95.0	93.5	16
0.75Bi _{0.85} Nd _{0.15} FeO ₃ -0.25BaTiO ₃	6.7	3.6 _{125 °C}	77.0	68.1 _{125 °C}	17
0.62BiFeO ₃ -0.30BaTiO ₃ - 0.08Nd(Zn _{0.5} Zr _{0.5})O ₃	10.5	2.4	87	63.9	18
Sm _{0.05} Ag _{0.85} (Nb _{0.7} Ta _{0.3})O ₃	14.0	7.0 _{120 °C}	85.0	94.5 _{120 °C}	19
0.40(Bi _{0.5} Na _{0.5} TiO ₃)-0.60(0.87BaTiO ₃ - 0.13Bi(Zn _{2/3} (Nb _{0.85} Ta _{0.15}) _{1/3})O ₃)	14.5	4.6	84.9	89.3	20
0.57BiFeO ₃ -0.30BaTiO ₃ - 0.13Bi(Li _{0.5} Nb _{0.5})O ₃	13.8	4.7 _{100 °C}	81.0	66.0 _{100 °C}	21
BNKT-20SSN ceramic (RRP)					
0.80Bi_{0.5}(Na_{0.82}K_{0.18})_{0.5}TiO₃- 0.20Sr(Sc_{0.5}Nb_{0.5})O₃	9.2	8.2	96.3	95.0	This work

Regarding the choice of references, it is standard practice to cite works from renowned journals and related systems to establish the scientific context and validity of our research. Such comparisons allow us to demonstrate the **novelty and relevance of our contributions** within the existing body of knowledge.

The references are as follows:

1. Li, D. et al. Improved energy storage properties achieved in (K, Na)NbO₃-based relaxor ferroelectric ceramics via a combinatorial optimization strategy. *Adv. Funct. Mater.* **32**, 2111776 (2021).
2. Luo, N. et al. Constructing phase boundary in AgNbO₃ antiferroelectrics: pathway simultaneously achieving high energy density and efficiency. *Nat. Commun.* **11**, 4824 (2020).
3. Chen, L. et al. Outstanding energy storage performance in high-hardness (Bi_{0.5}K_{0.5})TiO₃-

based lead-free relaxors via multi-scale synergistic design. *Adv. Funct. Mater.* **32**, 2110478 (2021).

4. Chen, L. et al. Giant energy-storage density with ultrahigh efficiency in lead-free relaxors via high-entropy design. *Nat. Commun.* **13**, 3089 (2022).

5. Hu, Q. et al. Achieve ultrahigh energy storage performance in BaTiO₃-Bi(Mg_{1/2}Ti_{1/2})O₃ relaxor ferroelectric ceramics via nano-scale polarization mismatch and reconstruction. *Nano Energy* **67**, 104264 (2020).

6. Zhang, M. et al. Significant increase in comprehensive energy storage performance of potassium sodium niobate-based ceramics via synergistic optimization strategy. *Energy Storage Mater.* **45**, 861-868 (2022).

7. Zhu, X. et al. Ultrahigh energy storage density in (Bi_{0.5}Na_{0.5})_{0.65}Sr_{0.35}TiO₃-based lead-free relaxor ceramics with excellent temperature stability. *Nano Energy* **98** 107276 (2022).

8. Li, D. et al. Lead-Free relaxor ferroelectric ceramics with ultrahigh energy storage densities via polymorphic polar nanoregions design. *Small* **19**, 2206958 (2022).

9. Yan, F. et al. Gradient-structured ceramics with high energy storage performance and excellent stability. *Small* **19**, 2206125 (2023).

10. Yang, W. et al. Superior energy storage properties in NaNbO₃-based ceramics via synergistically optimizing domain and band structures. *J. Mater. Chem. A* **10**, 11613-11624 (2022).

11. Qi, H. et al. Superior energy-storage capacitors with simultaneously giant energy density and efficiency using nanodomain engineered BiFeO₃-BaTiO₃-NaNbO₃ lead-free bulk ferroelectrics. *Adv. Energy. Mater.* **10**, 1903338 (2019).

12. Yuan, Q. et al. Bioinspired hierarchically structured all-inorganic nanocomposites with significantly improved capacitive performance. *Adv. Funct. Mater.* **30**, 2000191 (2020).

13. Xie, A. et al. NaNbO₃-(Bi_{0.5}Li_{0.5})TiO₃ lead-free relaxor ferroelectric capacitors with superior energy-storage performances via multiple synergistic design. *Adv. Energy. Mater.* **11**, 2101378 (2021).

14. Li, J. et al. Grain-orientation-engineered multilayer ceramic capacitors for energy storage applications. *Nat. Mater.* **19**, 999-1005 (2020).

15. Li, J. et al. Multilayer lead-free ceramic capacitors with ultrahigh energy density and efficiency. *Adv. Mater.* **30**, 1802155 (2018).

16. Cai, Z. et al. High-temperature lead-free multilayer ceramic capacitors with ultrahigh

energy density and efficiency fabricated via two-step sintering. *J. Mater. Chem. A* **7**, 14575-14582 (2019).

17. Wang, D. et al. Bismuth ferrite-based lead-free ceramics and multilayers with high recoverable energy density. *J. Mater. Chem. A* **6**, 4133-4144 (2018).

18. Wang, G. et al. Ultrahigh energy storage density lead-free multilayers by controlled electrical homogeneity. *Energy Environ. Sci.* **12**, 582-588 (2019).

19. Zhu, L.F. et al. Heterovalent-doping-enabled atom-displacement fluctuation leads to ultrahigh energy-storage density in AgNbO₃-based multilayer capacitors. *Nat. Commun.* **14**, 1166 (2023).

20. Zhao, P. et al. Ultrahigh energy density with excellent thermal stability in lead-free multilayer ceramic capacitors via composite strategy design. *J. Mater. Chem. A* **9**, 25914-25921 (2021).

21. Wang, G. et al. Fatigue resistant lead-free multilayer ceramic capacitors with ultrahigh energy density. *J. Mater. Chem. A* **8**, 11414-11423 (2020).

3. I still think that the comparison in Fig.2i is meaningless, There is no basis for these set values, e.g. the W_{rec} is set to 10 J cm⁻³, W_d is set to 6 J cm⁻³, either in experimentally or for application.

Reply: Frankly, we are puzzled by the persistence with which you are adhering to this matter. First and foremost, the parameters (W_{tot} , W_{rec} , η , W_d , $W_{rec} \sim 150$ °C and $\eta \sim 150$ °C) in Fig. 2i are all important parameters relevant to the **core performance and practical applications** of ceramic dielectric materials for capacitors. Furthermore, the data points presented in the figure are meticulously curated from seminal works published in prestigious journals, showcasing the epitome of excellence in characterizing the comprehensive performance of diverse dielectric matrices. Rest assured, the selection of these parameters and data sources has been rigorously scrutinized to ensure the utmost relevance and validity. It is worth noting that similar data selection methodologies can be observed in recent research reports of comparable nature ^{1,2}.

Fig. 3 Comparisons of comprehensive properties (W_{rec} , η , P_D , W_D , temperature, and frequency stability) between BT-BNT-NN ceramics and some representative energy storage ceramics with excellent comprehensive performance of different systems ¹.

Fig. 4 Comparison of the essential parameters for various [001]C-oriented PbTiO_3 -based relaxor ferroelectric single crystals with MPB composition. The 0.06PSN-0.61PMN-0.33PT crystals cover the largest area, denoting a superior overall performance ².

In Reference 1 (Fig. 3), the values of W_{rec} and W_d are set to 12 J cm^{-3} and 4 J cm^{-3} , respectively, while in Reference 2 (Fig. 4), the value of d_{33} is assigned as 3000 pC/N . The choice of these specific values might have been made to achieve a balance between various design considerations, such as **optimizing performance and maintaining a visually pleasing image**. Increasing the value of d_{33} to 4000 pC/N or higher is possible, but it's important to note that doing so could lead to **potential**

challenges in terms of aesthetics. The objective of adopting this approach is to enhance the overall visual appeal of the image. However, if we increase the value of d_{33} too much, it might introduce additional white space or other visual distortions that could compromise the image's overall aesthetic quality.

Fig. 5 Comparisons of comprehensive properties (W_{tot} , W_{rec} , η , W_d , $W_{rec} \sim 150\text{ }^{\circ}\text{C}$ and $\eta \sim 150\text{ }^{\circ}\text{C}$) between our study and other representative ceramics with excellent energy storage comprehensive performance.

Fig. 6 Comparisons of comprehensive properties (W_{tot} , W_{rec} , η , W_d , $W_{rec} \sim 150\text{ }^{\circ}\text{C}$ and $\eta \sim 150\text{ }^{\circ}\text{C}$) between our study and other representative ceramics with excellent energy storage comprehensive performance.

To suit your preferences, we can certainly adjust parameters W_d and W_{rec} to values like 12 J cm^{-3}

³, 30 J cm⁻³, or even higher. However, it's crucial to carefully consider the impact on **aesthetics** and ensure that any modifications align with the intended visual representation while maintaining the desired performance characteristics, as demonstrated in Fig. 5 and 6.

The references are as follows:

1. Chen, L. et al. Local diverse polarization optimized comprehensive energy-storage performance in lead-free superparaelectrics. *Adv. Mater.* **34**, 2205787 (2022).
2. Yang, L. et al. Simultaneously achieving giant piezoelectricity and record coercive field enhancement in relaxor-based ferroelectric crystals. *Nat. Commun.* **13**, 2444 (2022).